# Optimistic Policy Optimization with General Function Approximations

## Abstract

Although policy optimization with neural networks has a track record of achieving state-of-the-art results in reinforcement learning on various domains, the theoretical understanding of the computational and sample efficiency of policy optimization remains restricted to linear function approximations with finite-dimensional feature representations, which hinders the design of principled, effective, and efficient algorithms. To this end, we propose an optimistic model-based policy optimization algorithm, which allows general function approximations while incorporating exploration. In the episodic setting, we establish a $\sqrt{T}$-regret that scales polynomially in the eluder dimension of the general model class. Here $T$ is the number of steps taken by the agent. In particular, we specialize such a regret to handle two nonparametric model classes; one based on reproducing kernel Hilbert spaces and another based on overparameterized neural networks.

## 1 Introduction

Reinforcement learning with neural networks achieved impressive empirical breakthroughs (Mnih et al., 2015; Silver et al., 2016; 2017; Berner et al., 2019; Vinyals et al., 2019). These algorithms are often based on policy optimization (Williams, 1992; Baxter & Bartlett, 2000; Sutton et al., 2000; Kakade, 2002; Schulman et al., 2015; 2017). Compared with value-based approaches, which iteratively estimate the optimal value function, policy-based approaches directly optimize the expected total reward, which leads to more steady policy improvement. In particular, as shown in this paper, policy optimization generates steadily improving stochastic policies and consequently allow adversarial environments.

On the other hand, policy optimization often suffers from a lack of computational and statistical efficiency in practice, which calls for the principled design of efficient algorithms. Specifically, in terms of computational efficiency, the recent progress (Abbasi-Yadkori et al., 2019a;b; Bhandari & Russo, 2019; Liu et al., 2019; Agarwal et al., 2019; Wang et al., 2019) establishes the convergence of policy optimization to a globally optimal policy given sufficiently many data points, even in the presence of neural networks. However, in terms of sample efficiency, it remains less understood how to sequentially acquire the data points used in policy optimization while balancing exploration and exploitation, especially in the presence of neural networks, despite the recent progress (Cai et al., 2019; Agarwal et al., 2020). In particular, such a lack of sample efficiency prohibits the principled applications of policy optimization in critical domains, e.g., autonomous driving and dynamic treatment, where data acquisition is expensive.

In this paper, we aim to provably achieve sample efficiency in model-based policy optimization, which is quantified via the lens of regret. In particular, we focus on the episodic setting with general function approximations on the transition kernel. Such a setting is studied by Russo & Van Roy (2013; 2014); Osband & Van Roy (2014); Ayoub et al. (2020); Wang et al. (2020), which however focus on value iteration. In contrast, policy optimization remains less understood, despite its critical role in practice. To this end, we propose an optimistic policy optimization algorithm, which achieves exploration by incorporating optimism into policy evaluation and propagating it through policy improvement. In particular, we establish a $\kappa(\mathcal{P}) \cdot \sqrt{H^3 T}$-regret of the proposed algorithm, which matches that of existing value iteration algorithms but additionally allow the reward function to adversarially vary across each episode. Here $T$ is the number of steps, $H$ is the length of each episode, and $\kappa(\mathcal{P})$ is

the model capacity, which is defined based on the eluder dimension. Moreover, we instantiate the proposed algorithm for the special cases of reproducing kernel Hilbert spaces and overparameterized neural networks, both of which are infinite-dimensional model classes.

Our work is related to the study on computational efficiency of policy optimization (Fazel et al., 2018; Yang et al., 2019; Abbasi-Yadkori et al., 2019a;b; Bhandari & Russo, 2019; Liu et al., 2019; Agarwal et al., 2019; Wang et al., 2019). These works assume either the transition model is known or there exists a well-explored behavior policy such that the policy update direction can be estimated accurately. With such assumptions, the tradeoff between exploration and exploitation is absent and their focus is solely on the computational aspect. In addition, our work is related to the works on adversarial MDP (Even-Dar et al., 2009; Yu et al., 2009; Neu et al., 2010a;b; Zimin & Neu, 2013; Neu et al., 2012; Rosenberg & Mansour, 2019b;a). The algorithm in these work directly estimates the visitation measure and their algorithm utilize mirror descent to handle adversarial reward functions. Furthermore, our work is closely related the recent work on the sample complexity of policy optimization methods Cai et al. (2019), which only focus on the tabular and linear settings. In contrast, our work consider general function approximation setting, which is significantly more general. Moreover, the construction of optimistic policy evaluation is related to Ayoub et al. (2020), where the similar approach is incorporated in estimating the optimal value function. The theoretical foundation of such a type of optimistic estimation is innovated by Russo & Van Roy (2014) in the bandit problem. In particular, to characterizing the optimism and accuracy of the optimistic evaluation, we rely on the notion of the eluder dimension proposed by Russo & Van Roy (2014), which is further instantiated by this paper to the cases of kernel and neural function approximations.

### 1.1 NOTATIONS

We denote by $\|\cdot\|_p$ the $\ell_p$-norm of a vector when $p \in \mathbb{N}$ or the spectral norm of a matrix when $p = 2$. For any two distributions $p_1, p_2$ over a discrete set $\mathcal{A}$, we denote by $D_{\mathrm{KL}}(p_1 \,\|\, p_2)$ the KL-divergence

$$D_{\mathrm{KL}}(p_1 \,\|\, p_2) = \sum_{a \in \mathcal{A}} p_1(a) \log \frac{p_1(a)}{p_2(a)}.$$

For any $a, b, x \in \mathbb{R}$, we define the clamp function

$$\mathrm{clamp}(x, a, b) = \begin{cases} b, & \text{if } x > b, \\ x, & \text{if } a \le x \le b, \\ a, & \text{if } x < a. \end{cases} \tag{1.1}$$

## 2 PRELIMINARIES

### 2.1 ONLINE REINFORCEMENT LEARNING WITH ADVERSARIAL REWARDS

We consider an episodic MDP $(\mathcal{S}, \mathcal{A}, H, \{P_h\}_{h=1}^H, \{r_h\}_{h=1}^H)$, where $\mathcal{S}$ is a continuous state space, $\mathcal{A}$ is a discrete action space, $H$ is the number of steps in each episode, $\{P_h\}_{h=1}^H$ represent the unknown transition model, and $\{r_h\}_{h=1}^H$ represent the reward function. In particular, for any $h \in [H]$, $P_h$ represents the transition kernel from a state-action pair $(s_h, a_h)$ at the $h$-th step to the next state $s_{h+1}$, while $r_h$ represents the reward function at the $h$-th step that maps a state-action pair to a deterministic reward. Moreover, we allow the reward function to vary across episodes and denote by $r_h^k$ the reward function at the $h$-th step of the $k$-th episode. In particular, $r_h^k$ depends on the trajectories before the $k$-th episode begins, possibly in an adversarial manner, and remains unobservable until the $k$-th episode ends. Without loss of generality, we assume each episode starts from a fixed state $s_1$ and all rewards fall in the interval $[0, 1]$.

For any $h \in [H]$, a policy $\pi_h$ represents the conditional distribution of the action given the state at the $h$-th step. We drop the subscript $h$ to represent the collection of policies at all steps and still refer to such a collection as a policy when it is clear from the context. For any $(k, h) \in \mathbb{N} \times [H]$, given a policy $\pi$ and reward functions $\{r_h^k\}_{h=1}^H$, the value function and $Q$-function at the $h$-th step of the

$k$-th episode are defined by

$$V_h^{\pi,k}(s) = \mathbb{E}_\pi\Big[\sum_{j=h}^{H} r_j^k(s_j, a_j) \,\Big|\, s_h = s\Big], \quad Q_h^{\pi,k}(s,a) = \mathbb{E}_\pi\Big[\sum_{j=h}^{H} r_j^k(s_j, a_j) \,\Big|\, s_h = s, a_h = a\Big]$$

for any $(s,a) \in \mathcal{S} \times \mathcal{A}$. Here the subscript $\pi$ in the expectation $\mathbb{E}_\pi[\cdot]$ denotes all actions are taken according to the policy $\pi$ except for the one given in the condition. An online algorithm aims to construct and execute a sequence of policies $\{\pi^k\}_{k \geq 1}$ and minimize the regret

$$\text{Regret}(T) = \max_\pi \sum_{k=1}^{K} \big(V_1^{\pi,k}(s_1) - V_1^{\pi^k,k}(s_1)\big), \tag{2.1}$$

where $K$ is the number of episodes and $T = KH$ is the number of steps taken by the algorithm.

## 2.2 Reproducing Kernel Hilbert Space

We say $\mathcal{H}$ is a reproducing kernel Hilbert space (RKHS) on a set $\mathcal{Y}$ with the reproducing kernel $\mathcal{K} : \mathcal{Y} \times \mathcal{Y} \to \mathbb{R}$ if there exists an inner product $\langle \cdot, \cdot \rangle_{\mathcal{H}}$ such that, for any $f \in \mathcal{H}$ and $x \in \mathcal{Y}$, we have $f(x) = \langle f, \mathcal{K}_x \rangle_{\mathcal{H}}$. Here $\mathcal{K}_x$ represents the function $\mathcal{K}(x, \cdot)$, which is the Riesz representation of the evaluation functional at $x$ (Schölkopf et al., 2002). When the reproducing kernel $\mathcal{K}$ is continuous, symmetric, and positive definite, Mercer's theorem (Steinwart & Christmann, 2008) says $\mathcal{K}$ has the representation

$$\mathcal{K}(x,y) = \sum_{j=1}^{\infty} \lambda_j \phi_j(x) \phi_j(y), \text{ for any } x, y \in \mathcal{Y}, \tag{2.2}$$

where $\{\phi_j\}_{j=1}^{\infty}$ is an orthonormal basis of $L_2(\mathcal{Y})$ and $\lambda_1 \geq \lambda_2 \geq \cdots \geq 0$. See more details on RKHS in Section A.

## 3 Algorithm

**Framework:** Before the $k$-th episode begins, we construct the policy $\pi^k$ based on $\pi^{k-1}$ and $\{Q_h^{k-1}\}_{h=1}^{H}$, which are the policy in the $(k-1)$-th episode and estimators of $\{Q_h^{\pi^{k-1},k-1}\}_{h=1}^{H}$, respectively. Then, we execute the policy $\pi^k$ in the $k$-th episode and correspondingly update the $Q$-function estimators $\{Q_h^k\}_{h=1}^{H}$ using the reward function $\{r_h^k\}_{h=1}^{H}$, which is observed after the $k$-th episode ends.

**Policy Improvement:** For any $(k,h) \in [K] \times [H]$, we parametrize $\pi_h^k$ by

$$\pi_h^k(a \,|\, s) = \frac{\exp\{E_h^k(s,a)\}}{\sum_{a' \in \mathcal{A}} \exp\{E_h^k(s,a')\}}, \text{ for any } (s,a) \in \mathcal{S} \times \mathcal{A}. \tag{3.1}$$

Here $E_h^k$ is the potential function, which is initialized as the zero function and updated by

$$E_h^k(s,a) = E_h^{k-1}(s,a) + \alpha \cdot Q_h^{k-1}(s,a). \tag{3.2}$$

Here $\alpha > 0$ is the stepsize of policy improvement. Equivalently, we have

$$\pi_h^k(\cdot \,|\, s) \propto \pi_h^{k-1}(\cdot \,|\, s) \cdot \exp\big(\alpha \cdot Q_h^{k-1}(s, \cdot)\big)$$

for any $s \in \mathcal{S}$. To see (3.2) is a policy improvement step, note that $\pi_h^k$ is the maximizer of

$$L_h^k(\pi_h) = \mathbb{E}_{\pi^{k-1}}\big[\langle Q_h^{k-1}(s_h, \cdot), \pi_h(\cdot \,|\, s_h)\rangle_{\mathcal{A}} + \alpha^{-1} \cdot D_{\text{KL}}\big(\pi_h(\cdot \,|\, s_h) \big\| \pi_h^{k-1}(\cdot \,|\, s_h)\big)\big].$$

This is the same as the update in Politex (Abbasi-Yadkori et al., 2019a), which originates at MDP-E (Even-Dar et al., 2009). It is also very close to a slightly changed variant of the one-step objective in the proximal policy optimization (PPO) algorithm (Schulman et al., 2017; 2015).

**Policy Evaluation:** Let $\mathcal{P}$ be a known class of transition models such that $P_h \in \mathcal{P}$ for any $h \in [H]$, which is specified in Section 4. Also, for any $P \in \mathcal{P}$, $s \in \mathcal{S}$, $a \in \mathcal{A}$, and $V : \mathcal{S} \to [0, H]$, we define

$$z_P(s, a, V) = \int_{\mathcal{S}} V(s') \cdot P(s' \,|\, s, a) \, \mathrm{d}s'. \tag{3.3}$$

For any $(k, h) \in [K] \times [H]$, we construct a confidence set of the transition model $P_h$ and correspondingly the optimistic $Q$-function estimator $Q_h^k$ using the data collected before the $k$-th episode begins. Note that we do not use the data collected from the $k$-th episode although they are available, which, however, is only used to simplify the analysis. Let $V_{H+1}^k$ be the zero function. Inspired by Ayoub et al. (2020), given the optimistic value function estimators $\{V_{h+1}^\tau\}_{\tau=1}^{k-1}$ from the first $(k-1)$ episodes, we construct the confidence set $\mathcal{P}_h^k$ of $P_h$ by

$$\mathcal{P}_h^k = \left\{ P \in \mathcal{P} \,\Big|\, \sum_{\tau=1}^{k-1} \big(z_P(s_h^\tau, a_h^\tau, V_{h+1}^\tau) - z_{P_h^k}(s_h^\tau, a_h^\tau, V_{h+1}^\tau)\big)^2 \le \beta \right\}, \tag{3.4}$$

$$\text{where } P_h^k = \underset{P \in \mathcal{P}}{\operatorname{argmin}} \sum_{\tau=1}^{k-1} \big(V_{h+1}^\tau(s_{h+1}^\tau) - z_P(s_h^\tau, a_h^\tau, V_{h+1}^\tau)\big)^2$$

for a threshold $\beta > 0$, which represents the degree of optimism. Then, for any $(s, a) \in \mathcal{S} \times \mathcal{A}$, given the optimistic value function estimator $V_{h+1}^k$, we define the optimistic $Q$-function estimator $Q_h^k$ by

$$Q_h^k(s, a) = r_h^k(s, a) + \max_{P \in \mathcal{P}_h^k} z_P(s, a, V_{h+1}^k) \tag{3.5}$$

and correspondingly update the optimistic value function estimator by $V_h^k(s) = \langle Q_h^k(s, \cdot), \pi_h^k(\cdot \,|\, s) \rangle_{\mathcal{A}}$ for any $s \in \mathcal{S}$. We apply the clamp function defined in (1.1) to the second term on the right-hand side of (3.5) to ensure it falls in the range $[0, H - h]$, which is due to the assumption that all rewards fall in the range $[0, 1]$.

**Implementation:** The full algorithm is presented in Algorithm 1. Given a parametrization of the model class $\mathcal{P}$, we can apply the projected stochastic gradient descent (PSGD) algorithm to solve the constrained minimization problem in Line 11 of Algorithm 1. In particular, for kernel function approximations in Section 4.1, it reduces to a convex optimization problem, which allows the PSGD algorithm to converge to a global minimizer. Meanwhile, for neural function approximations, it reduces to an approximately convex problem in the overparametrized regime (Arora et al., 2019), which leads to the same global convergence guarantee. Also, to implement Lines 12 and 13 of Algorithm 1, it suffices to solve a constrained maximization problem (Feng et al., 2020), where the constraint is defined in Line 12. The Lagrangian relaxation of such a constrained maximization problem can be solved by the PSGD algorithm in the same manner of Line 11. In addition, to instantiate the update of $Q_h^k$ in Line 13, it suffices to solve a least-squares regression problem. In summary, we can instantiate the aforementioned steps through supervised learning oracles, which can be implemented in a computationally efficient manner.

## 4    THEORY

We analyze the regret of Algorithm 1, which is defined in (2.1). In Sections 4.1 and 4.2, we characterize the regret with specific choices of the model class $\mathcal{P}$, while in Section 4.3, we characterize the regret for a general $\mathcal{P}$, which serves as a meta result. An informal version of the results is given in the following theorem.

---

**Algorithm 1** Optimistic Policy Optimization with General Function Approximations

---

1: **Input:** number of episodes $K$, model class $\mathcal{P}$, stepsize $\alpha$, threshold $\beta$
2: Initialize $\pi^0$ as the uniformly random policy
3: **For** $k = 1$ to $K$ **do**
4:     Start the $k$-th episode and receive the initial state $s_1^k$
5:     **For** step $h = 1$ to $H$ **do**                                        *(policy improvement)*
6:         Update the policy by $\pi_h^k(\cdot \mid s) \propto \pi_h^{k-1}(\cdot \mid s) \exp\{\alpha Q_h^{k-1}(s, \cdot)\}$ for any $s \in \mathcal{S}$
7:         Take the action $a_h^k \sim \pi_h^k(\cdot \mid s_h^k)$ and receive the next state $s_{h+1}^k$
8:     Observe the reward function $\{r_h^k(\cdot, \cdot)\}_{h=1}^H$
9:     Initialize $V_{H+1}^k(\cdot)$ as the zero function
10:     **For** step $h = H$ to $1$ **do**                                        *(policy evaluation)*
11:         $P_h^k \leftarrow \operatorname{argmin}_{P \in \mathcal{P}} \sum_{\tau=1}^{k-1} (V_{h+1}^\tau(s_{h+1}^\tau) - \int_{\mathcal{S}} V_{h+1}^\tau(s') P(s' \mid s_h^\tau, a_h^\tau) \, \mathrm{d}s')^2$
12:

$$\mathcal{P}_h^k \leftarrow \{P \in \mathcal{P} : \sum_{\tau=1}^{k-1} (\int_{\mathcal{S}} V_{h+1}^\tau(s') P(s' \mid s_h^\tau, a_h^\tau) \, \mathrm{d}s' - \int_{\mathcal{S}} V_{h+1}^\tau(s') P_h^k(s' \mid s_h^\tau, a_h^\tau) \, \mathrm{d}s')^2 \leq \beta\}$$

13:         $Q_h^k(\cdot, \cdot) \leftarrow r_h^k(\cdot, \cdot) + \operatorname{clamp}(\max_{P \in \mathcal{P}_h^k} \int_{\mathcal{S}} V_{h+1}^k(s') P(s' \mid \cdot, \cdot) \, \mathrm{d}s', 0, H - h)$
14:         $V_h^k(\cdot) \leftarrow \langle Q_h^k(\cdot, \cdot), \pi_h^k(\cdot \mid \cdot) \rangle_{\mathcal{A}}$

---

**Theorem 4.1** (Informal Version of Theorems 4.3, 4.7, and 4.12)**.** With proper choices of $\alpha$ and $\beta$, the regret of Algorithm 1 satisfies

$$\operatorname{Regret}(T) = \widetilde{O}\big(\kappa(\mathcal{P}) \cdot \sqrt{H^3 T}\big)$$

with high probability. Here $\widetilde{O}$ omits absolute constants and logarithmic factors of $H$, $T$, and $|\mathcal{A}|$, while $\kappa(\mathcal{P})$ denotes the model capacity of $\mathcal{P}$, which is specified in Sections 4.1, 4.2, and 4.3.

Theorem 4.1 indicates that, compared with the optimal policy in hindsight, namely $\operatorname{argmax}_\pi \sum_{k=1}^K V_1^{\pi,k}(s_1)$, the average regret of Algorithm 1, namely $\operatorname{Regret}(T)/T$, converges to zero at a sublinear rate. In other words, at least one of the $K$ policies attained by Algorithm 1 achieves a vanishing optimality gap with respect to the varying reward function across the $K$ episodes. The model capacity $\kappa(\mathcal{P})$ is specified in Sections 4.1 and 4.2 for kernel and neural function approximations, respectively. To establish such specific results, we characterize $\kappa(\mathcal{P})$ using the eluder dimension for general function approximations in Section 4.3, which serves as the unified analysis.

### 4.1 KERNEL FUNCTION APPROXIMATIONS

Let $\mathcal{P}$ be a subset of an RKHS $\mathcal{H}$ with the reproducing kernel $\mathcal{K}$, which has the representation in (2.2). In detail, let $\mathcal{S}$ be a measurable set with $|\mathcal{S}| \leq 1$, where $|\cdot|$ denotes the Lebesgue measure. With a slight abuse of notation, we denote by $\mathcal{A}$ the embedding of the action space into a Euclidean space with the dimension $|\mathcal{A}|$, where $|\mathcal{A}|$ denotes the number of actions. Meanwhile, let $\mathcal{Y}$ be a $d_{\mathcal{Y}}$-dimensional set such that $\mathcal{S} \times \mathcal{A} \times \mathcal{S} \subset \mathcal{Y}$. We assume there exists $R \geq 2$ such that $\mathcal{P} \subset \mathcal{H}_R$, where $\mathcal{H}_R$ is the RKHS ball over $\mathcal{Y}$ with the radius $R$.

**Assumption 4.2.** We assume $\mathcal{K}$ satisfies the following regularity conditions.

(i). It holds that $|\mathcal{K}(x, y)| \leq 1$, $|\phi_j(x)| \leq 1$, and $\lambda_j \leq 1$ for any $x, y \in \mathcal{Y}$ and $j \in \mathbb{N}$.

(ii). There exist a threshold $\gamma \in (0, 1/2)$ and absolute constants $C_1, C_2 > 0$ such that $\lambda_j \leq C_1 \cdot \exp(-C_2 j^\gamma)$ for any $j \in \mathbb{N}$.

Note that we can replace the 1's in the upper bounds of Assumption 4.2 with any absolute constant, which is reflected in the $\mathcal{H}$-norm of any function in $\mathcal{H}$. Meanwhile, we can relax $|\phi_j(x)| \leq 1$ into $|\lambda_j^\tau \cdot \phi_j(x)| \leq 1$ for any absolute constant $\tau \in [0, 1/2)$, which leads to the same regret.

We have the following result on the regret of Algorithm 1.

**Theorem 4.3.** Suppose Assumption 4.2 holds and $\mathcal{P} \subset \mathcal{H}_R$. There exist absolute constants $C_3, C_4 > 0$ such that, for any $p \in (0, 1)$, if we set

$$\alpha = \sqrt{2 \log |\mathcal{A}|/(HT)}, \quad \beta = C_3 H^2 \cdot \log^{1+1/\gamma}(RT/p) \cdot \log^2(1/\gamma)/\gamma$$

in Algorithm 1, then it holds that

$$\text{Regret}(T) \leq C_4 \sqrt{H^3 T} \cdot \log^{1+1/\gamma}(|\mathcal{A}|RT/p) \cdot \log^2(1/\gamma)/\gamma$$

with probability at least $(1 - p)$.

*Proof.* See Section C for a detailed proof. □

Theorem 4.3 indicates that $\log^{1+1/\gamma}(|\mathcal{A}|RT/p) \cdot \log^2(1/\gamma)/\gamma$ serves as the model capacity $\kappa(\mathcal{P})$ in Theorem 4.1 for kernel function approximations. In particular, we can obtain $\gamma$ for a broad range of reproducing kernels $\mathcal{K}$ (Srinivas et al., 2009). Meanwhile, we can scale $R$ to control the model capacity $\kappa(\mathcal{P})$ (Schölkopf et al., 2002).

## 4.2 NEURAL FUNCTION APPROXIMATIONS

Let $\mathcal{P}$ be a set of overparametrized neural networks. In detail, we denote by NN a neural network with its weights collected in a vector $w \in \mathbb{R}^m$. Let $w^0$ be the random initial weights. For a radius $R \geq 2$, we define

$$\mathcal{P} = \{P : \exists w \in \mathcal{B}_R, \text{ s.t. } P(s' \mid s, a) = \text{NN}(x; w), \text{ for any } x = (s, a, s') \in \mathcal{S} \times \mathcal{A} \times \mathcal{S} \subset \mathcal{Y}\},$$

$$\text{where } \mathcal{B}_R = \{w \in \mathbb{R}^m : \|w - w^0\|_2 \leq R\}. \tag{4.1}$$

Without loss of generality, we assume $\text{NN}(x; w^0) = 0$ for any $x \in \mathcal{Y}$, which can be achieved by a symmetric initialization scheme. See Section E for a detailed explanation. To connect with the result for kernel function approximations in Section 4.1, we define the following condition.

**Condition 4.4** (Implicit Linearization)**.** It holds that

$$\xi_m = \max_{x \in \mathcal{Y}, w \in \mathcal{B}_R} |\text{NN}(x; w) - \nabla_w \text{NN}(x; w^0)^\top (w - w^0)| \leq 1/(4K^{3/2}H).$$

Condition 4.4 indicates that $\text{NN}(x; w)$ is uniformly close to the linear function $\nabla_w \text{NN}(x; w^0)^\top (w - w^0)$ of $w$. In particular, the linearization error $\xi_m$ is negligible compared with the dominating terms in the regret. The following lemma ensures Condition 4.4 holds for two-layer neural networks when $m$ is sufficiently large.

**Lemma 4.5** (Overparametrization)**.** Suppose NN is a two-layer neural network, where the activation function is 1-smooth, and it holds that $\|x\|_2 \leq 1$ for any $x \in \mathcal{Y}$. Then, Condition 4.4 holds when $m \geq d_{\mathcal{Y}} R^4 K^3 H^2$.

*Proof.* See Section E for a detailed proof. □

Note that the analogous of Lemma 4.5 also applies to nonsmooth activation functions, for example, the rectified linear unit (ReLU), and multilayer neural networks (Allen-Zhu et al., 2019; Du et al., 2019; Zou et al., 2020; Gao et al., 2019), which ensures Condition 4.4 holds. The linear function of $w$ in Condition 4.4 induces an RKHS $\mathcal{H}$ with the reproducing kernel

$$\mathcal{K}_{\text{NTK}}(x, y) = \nabla_w \text{NN}(x; w^0)^\top \nabla_w \text{NN}(y; w^0), \text{ for any } x, y \in \mathcal{Y}, \tag{4.2}$$

which is known as the neural tangent kernel (NTK) (Jacot et al., 2018).

**Assumption 4.6.** We assume $\mathcal{K}_{\text{NTK}}$ satisfies the regularity conditions in Assumption 4.2.

Note that the NTK defined in (4.2) depends on the randomness of $w^0$. When $m$ goes to infinity, such an empirical NTK converges to its expectation, which gives the population NTK. It is shown in Yang

& Salman (2019) that Assumption 4.2 holds for the population NTK, which implies it also holds for the empirical NTK with high probability when $m$ is sufficiently large.

We have the following result on the regret of Algorithm 1.

**Theorem 4.7.** Suppose Assumption 4.6 and Condition 4.4 hold and $\mathcal{P}$ has the representation in (4.1). There exist absolute constants $C_5, C_6 > 0$ such that for any $p \in (0, 1)$, if we set

$$\alpha = \sqrt{2 \log |\mathcal{A}|/(HT)}, \quad \beta = C_5 H^2 \cdot \log^{1+1/\gamma}(RT/p) \cdot \log^2(1/\gamma)/\gamma$$

in Algorithm 1, then it holds that

$$\text{Regret}(T) \leq C_6 \sqrt{H^3 T} \cdot \log^{1+1/\gamma}(|\mathcal{A}|RT/p) \cdot \log^2(1/\gamma)/\gamma$$

with probability at least $(1-p)$.

*Proof.* See Section D for a detailed proof. □

In parallel with Theorem 4.3, Theorem 4.7 indicates that $\log^{1+1/\gamma}(|\mathcal{A}|RT/p) \cdot \log^2(1/\gamma)/\gamma$ serves as the model capacity $\kappa(\mathcal{P})$ in Theorem 4.1 for neural function approximations, which can be controlled by scaling $R$ (Arora et al., 2019).

### 4.3 General Function Approximations

Let $\mathcal{P}$ be a general model class, whose model capacity is characterized by the eluder dimension (Russo & Van Roy, 2014; Osband & Van Roy, 2014; Ayoub et al., 2020) defined as follows.

**Definition 4.8** (Eluder Dimension). Let $\mathcal{Z}$ be a set of real-valued functions on the domain $\mathcal{X}$. For any $\varepsilon > 0$ and $\tau \in \mathbb{N}$, we say $x_\tau \in \mathcal{X}$ is $(\mathcal{Z}, \varepsilon)$-independent of $x_1, \ldots, x_{\tau-1} \in \mathcal{X}$ if there exist $f_1, f_2 \in \mathcal{Z}$ such that

$$\left(\sum_{j=1}^{\tau-1} |f_1(x_j) - f_2(x_j)|^2\right)^{1/2} \leq \varepsilon, \quad |f_1(x_\tau) - f_2(x_\tau)| > \varepsilon. \tag{4.3}$$

The eluder dimension of $\mathcal{Z}$ at scale $\varepsilon$, which is denoted by $\dim_{\mathrm{E}}(\mathcal{Z}, \varepsilon)$, is the length of the longest sequence $x_1, \ldots, x_\tau \in \mathcal{X}$ such that, for any $j \in [\tau]$, $x_j$ is $(\mathcal{Z}, \varepsilon')$-independent of $x_1, \ldots, x_{j-1}$ for some $\varepsilon' \geq \varepsilon$.

The following lemma decomposes the regret of Algorithm 1 into errors that arise from policy improvement and policy evaluation, respectively.

**Lemma 4.9** (Regret Decomposition). For any $k \in [K]$, it holds that

$$\text{Regret}(T) = \sum_{k=1}^{K}\sum_{h=1}^{H} \mathbb{E}_{\pi^*}[\langle Q_h^k(\cdot, s_h), \pi_h^*(\cdot \mid s_h) - \pi_h^k(\cdot \mid s_h)\rangle] + \sum_{k=1}^{K}\left(V_1^k(s_1) - V_1^{\pi^k, k}(s_1)\right)$$
$$+ \sum_{k=1}^{K}\sum_{h=1}^{H} \mathbb{E}_{\pi^*}[r_h^k(s_h, a_h) + z_{P_h}(s_h, a_h, V_{h+1}^k) - Q_h^k(s_h, a_h)].$$

*Proof.* See Lemma 4.2 of Cai et al. (2019) for a detailed proof. □

The following lemma characterizes the error that arises from policy improvement.

**Lemma 4.10** (Policy Improvement). It we set $\alpha = \sqrt{2 \log |\mathcal{A}|/(KH^2)}$ in Algorithm 1, then it holds that

$$\sum_{k=1}^{K}\sum_{h=1}^{H} \mathbb{E}_{\pi^*}[\langle Q_h^k(\cdot, s_h), \pi_h^*(\cdot \mid s_h) - \pi_h^k(\cdot \mid s_h)\rangle] \leq \sqrt{2KH^4 \cdot \log |\mathcal{A}|}.$$

*Proof.* See Section B.2 for a detailed proof. □

Recall that for any $P \in \mathcal{P}$, $z_P$ is defined in (3.3). Also, let $\mathcal{Z}_\mathcal{P} = \{z_P : P \in \mathcal{P}\}$. For any $\epsilon > 0$, we denote by $\mathcal{N}_\epsilon(\mathcal{P}, \|\cdot\|_{\infty,1})$ the $\epsilon$-covering number of $\mathcal{P}$ with respect to the $\ell_{\infty,1}$-norm distance, which is defined by

$$\|P - P'\|_{\infty,1} = \max_{(s,a) \in \mathcal{S} \times \mathcal{A}} \int_\mathcal{S} |P(s' \,|\, s, a) - P'(s' \,|\, s, a)| \,\mathrm{d}s', \text{ for any } P, P' \in \mathcal{P}.$$

The following lemma characterizes the error that arises from policy evaluation.

**Lemma 4.11** (Policy Evaluation). For any $p \in (0, 1)$, if we set

$$\beta \geq 2H^2 \cdot \log\big(\mathcal{N}_{1/(KH)}(\mathcal{P}, \|\cdot\|_{\infty,1}) \cdot 2H/p\big) + 4\big(H + \sqrt{H^2/4 \cdot \log(8K^2 H/p)}\big) \quad (4.4)$$

in Algorithm 1, then the following results hold with probability at least $(1 - p)$.

- (Optimism) For any $(k, h) \in [K] \times [H]$ and $(s, a) \in \mathcal{S} \times \mathcal{A}$, it holds that

$$r_h^k(s, a) + z_{P_h}(s, a, V_{h+1}^k) - Q_h^k(s, a) \leq 0.$$

- (Accuracy) Let $d = K \wedge \dim_\mathrm{E}(\mathcal{Z}_\mathcal{P}, 1/K)$. It holds that

$$\sum_{k=1}^K \big(V_1^k(s_1) - V_1^{\pi^k,k}(s_1)\big) \leq \sqrt{32KH^3 \cdot \log(p/2)} + H(dH + 1) + 4\sqrt{d\beta KH^2}.$$

*Proof.* See Section B.1 for a detailed proof. □

Recall that $T = KH$. The following theorem characterizes the regret of Algorithm 1 when $\mathcal{P}$ is a general model class, which serves as a meta result.

**Theorem 4.12.** In Algorithm 1, if we set $\alpha$ as in Lemma 4.10 and $\beta$ as in Lemma 4.11, then it holds that

$$\mathrm{Regret}(T) \leq \sqrt{2H^3 T \cdot \log|\mathcal{A}|} + \sqrt{32H^2 T \cdot \log(p/2)} + H(dH + 1) + 4\sqrt{d\beta HT}$$

with probability at least $(1 - p)$, where $d = K \wedge \dim_\mathrm{E}(\mathcal{Z}_\mathcal{P}, 1/K)$.

*Proof.* The proof follows from combining Lemmas 4.9, 4.11, and 4.10. □

Theorem 4.12 indicates that

$$\max\big\{d, \sqrt{d \cdot \log\big(\mathcal{N}_{1/(KH)}(\mathcal{P}, \|\cdot\|_{\infty,1})\big)}\big\}$$

serves as the model capacity $\kappa(\mathcal{P})$ in Theorem 4.1. The regret upper bound in Theorem 4.12 is similar to that in Ayoub et al. (2020) when $\mathcal{P}$ is a general model class, whose model capacity is characterized by the eluder dimension. In contrast, our algorithm additionally handles adversarial rewards, which is a benefit of the policy optimization approach. To establish the regret upper bounds in Sections 4.1 and 4.2, it remains to characterize the corresponding eluder dimension and log-covering number, respectively. See Sections C and D for details. As a special case, Theorem 4.12 also applies to the case where $\mathcal{P}$ is a set of $\underline{d}$-dimensional linear models with a finite $\underline{d}$, which is studied in Cai et al. (2019). In particular, the eluder dimension and log-covering number in (4.4) are both $\widetilde{O}(\underline{d})$ (Ayoub et al., 2020), which leads to the $\sqrt{\underline{d}^2 H^3 T}$-regret in Cai et al. (2019). In contrast, Theorem 4.12 additionally handles the case where $\underline{d}$ is infinite as in kernel and neural function approximations.

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

## A    MORE DETAILS ON RKHS

An example of the RKHS is the linear model class. In particular, let $\psi$ be a $\underline{d}$-dimensional feature vector

$$\psi(x) = \big(\psi_1(x), \dots, \psi_{\underline{d}}(x)\big)^\top, \text{ for any } x \in \mathcal{Y},$$

where $\psi_1, \dots, \psi_{\underline{d}}$ are linearly independent. Then, the linear span of $\{\psi_j\}_{j=1}^\infty$ forms an RKHS $\mathcal{H}_\psi$ with the reproducing kernel $\mathcal{K}_\psi(x,y) = \psi(x)^\top \psi(y)$ for any $x, y \in \mathcal{Y}$, and the corresponding inner product $\langle \cdot, \cdot \rangle_{\mathcal{H}_\psi}$ is defined by the Euclidean inner product

$$\langle \psi(\cdot)^\top c_1, \psi(\cdot)^\top c_2 \rangle_{\mathcal{H}_\psi} = c_1^\top c_2$$

for any $c_1, c_2 \in \mathbb{R}^{\underline{d}}$.

The above example can be naturally generalized to the case where $\underline{d} = \infty$, that is, the feature vector can be infinite-dimensional. Moreover, recall that (2.2) says the reproducing kernel $\mathcal{K}$ has the representation

$$\mathcal{K}(x,y) = \sum_{j=1}^\infty \lambda_j \phi_j(x)\phi_j(y), \text{ for any } x, y \in \mathcal{Y},$$

where $\{\phi_j\}_{j=1}^\infty$ is an orthonormal basis of $L_2(\mathcal{Y})$ and $\lambda_1 \geq \lambda_2 \geq \cdots \geq 0$. We refer to $\{\phi_j\}_{j=1}^\infty$ as the eigenfunctions of $\mathcal{K}$ with the corresponding eigenvalues $\{\lambda_j\}_{j=1}^\infty$. Such a representation gives the feature vector

$$\widetilde{\phi}(x) = \big(\sqrt{\lambda_1} \cdot \phi_1(x), \sqrt{\lambda_2} \cdot \phi_2(x), \dots\big)^\top, \text{ for any } x \in \mathcal{Y}.$$

The linear span of $\{\sqrt{\lambda_j} \cdot \phi_j\}_{j=1}^\infty$ recovers the RKHS $\mathcal{H}$ with the reproducing kernel $\mathcal{K}$ and inner product $\langle \cdot, \cdot \rangle_{\mathcal{H}}$. When the reproducing kernel $\mathcal{K}$ is infinite-dimensional, that is, $\mathcal{K}$ has an infinite number of non zero eigenvalues, $\widetilde{\phi}$ is an infinite-dimensional vector. It is known that RKHSs of various infinite-dimensional reproducing kernels, for example, the Gaussian radius basis function kernel (Steinwart & Christmann, 2008), are rich function model classes in the sense that they are dense in the class of continuous and bounded functions.

## B    PROOFS FOR SECTION 4.3

In this section, we provide the detailed proofs of the result in Section 4.3. For notational simplicity, we denote by $\mathbb{P}_h$ the operator that takes the conditional expectation with respect to the transition kernel $P_h$ for any $h \in [H]$.

### B.1    PROOF OF LEMMA 4.11

*Proof.* We define the event $\mathcal{E}$ that

$$P_h \in \mathcal{P}_h^k, \text{ for any } (k,h) \in [K] \times [H]. \tag{B.1}$$

By our choice of $\beta$ and Lemma F.4 with $\delta = p/2$, it holds that $\mathcal{E}$ occurs with probability at least $(1 - p/2)$.

**Optimism:** For any $(k, h) \in [K] \times [H]$, $P_h \in \mathcal{P}_h^k$ implies

$$
\begin{aligned}
Q_h^k(\cdot, \cdot) &- \left( r_h^k(\cdot, \cdot) + (\mathbb{P}_h V_{h+1}^k)(\cdot, \cdot) \right) \\
&= \text{clamp} \left( \max_{P \in \mathcal{P}_h^k} \int_{\mathcal{S}} V_{h+1}^k(s') \cdot P(s' \,|\, \cdot, \cdot) \, \mathrm{d}s', 0, H - h \right) - \int_{\mathcal{S}} V_{h+1}^k(s') \cdot P_h(s' \,|\, \cdot, \cdot) \, \mathrm{d}s' \\
&\geq \text{clamp} \left( \int_{\mathcal{S}} V_{h+1}^k(s') \cdot P_h(s' \,|\, \cdot, \cdot) \, \mathrm{d}s', 0, H - h \right) - \int_{\mathcal{S}} V_{h+1}^k(s') \cdot P_h(s' \,|\, \cdot, \cdot) \, \mathrm{d}s'. \quad \text{(B.2)}
\end{aligned}
$$

When $h = H$, the right-hand side of (B.2) is zero since $V_{H+1}^k(\cdot) = 0$. When $h < H$, by the construction of $Q_{h+1}^k$ in (3.5) and the assumption that $r_{h+1}^k(\cdot) \in [0, 1]$, we have

$$
Q_{h+1}^k(\cdot, \cdot) \in [0, H - h], \quad V_{h+1}^k(\cdot) \in [0, H - h], \quad \int_{\mathcal{S}} V_{h+1}^k(s') \cdot P_h(s' \,|\, \cdot, \cdot) \, \mathrm{d}s' \in [0, H - h].
$$

The right-hand side of (B.2) is also zero, which implies

$$
Q_h^k(\cdot, \cdot) - \left( r_h^k(\cdot, \cdot) + (\mathbb{P}_h V_{h+1}^k)(\cdot, \cdot) \right) \geq 0.
$$

Thus, the optimism result holds under the event $\mathcal{E}$.

**Accuracy:** We invoke Lemma F.1 and obtain

$$
\sum_{k=1}^{K} V_1^k(s_1^k) - V_1^{\pi^k}(s_1^k) = \sum_{k=1}^{K} \sum_{h=1}^{H} (D_{h,1}^k + D_{h,2}^k) \quad \text{(B.3)}
$$
$$
+ \sum_{k=1}^{K} \sum_{h=1}^{H} \left( Q_h^k(s_h^k, a_h^k) - \left( r_h^k(s_h^k, a_h^k) + \mathbb{P}_h V_{h+1}^k(s_h^k, a_h^k) \right) \right),
$$

where $|D_{h,1}^k| \leq 2H$, $|D_{h,2}^k| \leq 2H$, $D_{H,2}^k = 0$ for any $(k, h) \in [K] \times [H]$, and

$$
\begin{aligned}
&D_{1,1}^1, D_{1,2}^1 + D_{2,1}^1, D_{2,2}^1 + D_{3,1}^1, \ldots, D_{H-1,2}^1 + D_{H,1}^1, \\
&D_{1,1}^2, D_{1,2}^2 + D_{2,1}^2, D_{2,2}^2 + D_{3,1}^2, \ldots, D_{H-1,2}^2 + D_{H,1}^2, \\
&\ldots \ldots
\end{aligned}
$$

is a martingale difference sequence. The Azuma-Hoeffding inequality (Azuma, 1967) implies

$$
\sum_{k=1}^{K} \sum_{h=1}^{H} (D_{h,1}^k + D_{h,2}^k) \leq \sqrt{32KH^3 \cdot \log(p/2)} \quad \text{(B.4)}
$$

with probability at least $(1 - p/2)$. It remains to upper bound the second term on the right-hand side of (B.3). For any $(k, h) \in [K] \times [H]$, $P_h \in \mathcal{P}_h^k$ implies

$$
\begin{aligned}
Q_h^k(s_h^k, a_h^k) &- \left( r_h^k(s_h^k, a_h^k) + (\mathbb{P}_h V_{h+1}^k)(s_h^k, a_h^k) \right) \\
&= \text{clamp} \left( \max_{P \in \mathcal{P}_h^k} \int_{\mathcal{S}} V_{h+1}^k(s') \cdot P(s' \,|\, s_h^k, a_h^k) \, \mathrm{d}s', 0, H - h \right) - \int_{\mathcal{S}} V_{h+1}^k(s') \cdot P_h(s' \,|\, s_h^k, a_h^k) \, \mathrm{d}s' \\
&\leq \max_{P \in \mathcal{P}_h^k} \int_{\mathcal{S}} V_{h+1}^k(s') \cdot P(s' \,|\, s_h^k, a_h^k) \, \mathrm{d}s' - \min_{P \in \mathcal{P}_h^k} \int_{\mathcal{S}} V_{h+1}^k(s') \cdot P(s' \,|\, s_h^k, a_h^k) \, \mathrm{d}s'.
\end{aligned}
$$

Applying Lemma F.5, for any $h \in [H]$, we have

$$Q_h^k(s_h^k, a_h^k) - \left(r_h^k(s_h^k, a_h^k) + (\mathbb{P}_h V_{h+1}^k)(s_h^k, a_h^k)\right)$$

$$\leq \sum_{k=1}^K \left(\max_{P \in \mathcal{P}_h^k} \int_{\mathcal{S}} V_{h+1}^k(s') \cdot P(s' \mid s_h^k, a_h^k) \, \mathrm{d}s' - \min_{P \in \mathcal{P}_h^k} \int_{\mathcal{S}} V_{h+1}^k(s') \cdot P(s' \mid s_h^k, a_h^k) \, \mathrm{d}s'\right)$$

$$\leq 1 + dH + 4\sqrt{d\beta K}, \tag{B.5}$$

where $d = K \wedge \dim_{\mathrm{E}}(\mathcal{Z}_{\mathcal{P}}, 1/K)$. Combining (B.3)-(B.5), we have

$$\sum_{k=1}^K V_1^k(s_1^k) - V_1^{\pi^k}(s_1^k) \leq \sqrt{32KH^3 \cdot \log(p/2)} + H + dH^2 + 4\sqrt{d\beta HT}.$$

In summary, when the event $\mathcal{E}$ and (B.4) hold, which occur with probability at least $(1-p)$, we have

$$Q_h^k(\cdot, \cdot) \geq r_h^k(\cdot, \cdot) + (\mathbb{P}_h V_{h+1}^k)(\cdot, \cdot), \text{ for any } (k, h) \in [K] \times [H],$$

$$\sum_{k=1}^K \left(V_1^k(s_1^k) - V_1^{\pi^k, k}(s_h^k)\right) \leq \sqrt{32KH^3 \cdot \log(p/2)} + H(1 + dH) + 4\sqrt{d\beta HT}.$$

Therefore, we conclude the proof of Lemma 4.11. □

### B.2 PROOF OF LEMMA 4.10

*Proof.* By Lemma 3.3 of Cai et al. (2019), for any $(k, h) \in [K] \times [H]$ and $s \in \mathcal{S}$, we have

$$\langle Q_h^k(s, \cdot), \pi_h^*(\cdot \mid s) - \pi_h^k(\cdot \mid s)\rangle$$

$$\leq \alpha H^2/2 + \alpha^{-1} \cdot \left(D_{\mathrm{KL}}\left(\pi_h^*(\cdot \mid s) \,\middle\|\, \pi_h^k(\cdot \mid s)\right) - D_{\mathrm{KL}}\left(\pi_h^*(\cdot \mid s) \,\middle\|\, \pi_h^{k+1}(\cdot \mid s)\right)\right), \tag{B.6}$$

which implies

$$\sum_{k=1}^K \sum_{h=1}^H \mathbb{E}_{\pi^*}[\langle Q_h^k(s_h, \cdot), \pi_h^*(\cdot \mid s_h) - \pi_h^k(\cdot \mid s_h)\rangle]$$

$$\leq \alpha K H^3/2 + \alpha^{-1} \cdot \sum_{h=1}^H \mathbb{E}_{\pi^*}\left[D_{\mathrm{KL}}\left(\pi_h^*(\cdot \mid s_h) \,\middle\|\, \pi_h^1(\cdot \mid s_h)\right)\right]$$

$$\leq \alpha K H^3/2 + \alpha^{-1} H \cdot \log|\mathcal{A}|. \tag{B.7}$$

Plugging $\alpha = \sqrt{2\log|\mathcal{A}|/(HT)}$ into the right-hand side of (B.7), we conclude the proof of Lemma 4.10. □

## C  PROOF OF THEOREM 4.3

In this section, we prove Theorem 4.3. By Theorem 4.12, it suffices to upper bound the eluder dimension of $\mathcal{Z}_{\mathcal{P}} = \{z_P : P \in \mathcal{P}\}$ and log-covering number of $\mathcal{H}_R$, which are characterized by the following two lemmas, respectively.

**Lemma C.1.** Under Assumption 4.2, there exists an absolute constant $C_7 > 0$ such that

$$K \wedge \dim_{\mathrm{E}}(\mathcal{Z}_{\mathcal{P}}, 1/K) \leq C_7 \cdot \log^2(1/\gamma)/\gamma \cdot \log^{1+1/\gamma}(RT).$$

*Proof.* Let $t = K \wedge \dim_{\mathrm{E}}(\mathcal{Z}_{\mathcal{P}}, 1/K)$. By Definition 4.8, there exists a sequence $x_1, \ldots, x_t \in \mathcal{S} \times \mathcal{A} \times [0, H]^{\mathcal{S}}$ such that $x_\tau$ is $(\mathcal{Z}_{\mathcal{P}}, 1/K)$-independent of $x_1, \ldots, x_{\tau-1}$ for any $\tau \in [t]$. Here the independency scale is assumed to be $1/K$ without loss of generality, which can be changed to any

value larger than $1/K$. In other words, for any $\tau \in [t]$, there exist $P_1, P_2 \in \mathcal{P}$ such that

$$\sum_{i=1}^{\tau-1} |z_{P_1}(x_i) - z_{P_2}(x_i)|^2 \leq 1/K^2, \tag{C.1}$$

$$|z_{P_1}(x_\tau) - z_{P_2}(x_\tau)| > 1/K. \tag{C.2}$$

Note that for any $x = (s, a, V) \in \mathcal{S} \times \mathcal{A} \times [0, H]^{\mathcal{S}}$ and $P \in \mathcal{P}$, by the reproducing property of the RKHS $\mathcal{H}$, we can write

$$z_P(x) = \int_{\mathcal{S}} V(s') \cdot P(s' \mid s, a) \, \mathrm{d}s' = \int_{\mathcal{S}} V(s') \cdot \langle P, \mathcal{K}_{(s,a,s')} \rangle_{\mathcal{H}} \, \mathrm{d}s'.$$

The representation of $\mathcal{K}$ in (2.2) implies

$$z_P(x) = \int_{\mathcal{S}} V(s') \cdot \Big\langle P, \sum_{j=1}^{\infty} \lambda_j \cdot \phi_j(s, a, s') \cdot \phi_j \Big\rangle_{\mathcal{H}} \, \mathrm{d}s'$$

$$= \int_{\mathcal{S}} V(s') \cdot \sum_{j=1}^{\infty} \sqrt{\lambda_j} \cdot \phi_j(s, a, s') \cdot \langle P, \sqrt{\lambda_j} \cdot \phi_j \rangle_{\mathcal{H}} \, \mathrm{d}s'$$

$$= \sum_{j=1}^{\infty} \sqrt{\lambda_j} \cdot \int_{\mathcal{S}} V(s') \cdot \phi_j(s, a, s') \, \mathrm{d}s' \cdot \langle P, \sqrt{\lambda_j} \cdot \phi_j \rangle_{\mathcal{H}}.$$

For any $d_0$ such that $d_0^\gamma \geq 4(1 - \gamma)(\gamma C_2)^{-1}$ where $\gamma$ and $C_2$ are defined in Assumption 4.2, we define

$$\widetilde{z}_P(x) = \sum_{1 \leq j \leq d_0} \sqrt{\lambda_j} \cdot \int_{\mathcal{S}} V(s') \cdot \phi_j(s, a, s') \, \mathrm{d}s' \cdot \langle P, \sqrt{\lambda_j} \cdot \phi_j \rangle_{\mathcal{H}}$$

for any $x = (s, a, V) \in \mathcal{S} \times \mathcal{A} \times [0, H]^{\mathcal{S}}$ and $P \in \mathcal{P}$. Then, we have

$$|z_P(x) - \widetilde{z}_P(x)| = \Big| \sum_{j > d_0} \sqrt{\lambda_j} \cdot \int_{\mathcal{S}} V(s') \cdot \phi_j(s, a, s') \, \mathrm{d}s' \cdot \langle P, \sqrt{\lambda_j} \cdot \phi_j \rangle_{\mathcal{H}} \Big|$$

$$\leq \sum_{j > d_0} \sqrt{\lambda_j} \cdot |\mathcal{S}| \cdot H \cdot \|P\|_{\mathcal{H}} \leq \sum_{j > d_0} \sqrt{\lambda_j} \cdot RH,$$

where $|\mathcal{S}|$ is the Lebesgue measure of $\mathcal{S}$ and $|\mathcal{S}| \leq 1$. Here the first inequality follows from the Cauchy-Schwarz inequality and $\|\sqrt{\lambda_j} \cdot \phi_j\|_{\mathcal{H}} = 1$ for any $j \in \mathbb{N}$, since $\{\sqrt{\lambda_j} \cdot \phi_j\}_{j=1}^{\infty}$ is an orthonormal basis of $\mathcal{H}$. By Assumption 4.2, we obtain

$$|z_P(x) - \widetilde{z}_P(x)| \leq \sum_{j > d_0} \sqrt{C_1} \cdot \exp(-C_2 j^\gamma / 2) \cdot RH$$

$$= \sqrt{C_1} \cdot RH \cdot \sum_{j > d_0} \exp(-C_2 j^\gamma / 2)$$

$$\leq \sqrt{C_1} \cdot RH \cdot 4 d_0^{1-\gamma} (\gamma C_2)^{-1} \cdot \exp(-C_2 d_0^\gamma / 2), \tag{C.3}$$

where the last inequality follows from Lemma F.6. For notational simplicity, let

$$\Gamma(d_0) = \sqrt{C_1} \cdot RH \cdot 4 d_0^{1-\gamma} (\gamma C_2)^{-1} \cdot \exp(-C_2 d_0^\gamma / 2). \tag{C.4}$$

By (C.1), it holds that

$$
\begin{aligned}
\sum_{i=1}^{\tau-1} |\widetilde{z}_{P_1}(x_i) - \widetilde{z}_{P_2}(x_i)|^2 &\le \sum_{i=1}^{\tau-1} \big(|z_{P_1}(x_i) - z_{P_2}(x_i)| + 2\Gamma(d_0)\big)^2 \\
&\le \sum_{i=1}^{\tau-1} \big(2|z_{P_1}(x_i) - z_{P_2}(x_i)|^2 + 8\Gamma(d_0)^2\big) \\
&\le 2/K^2 + 8t \cdot \Gamma(d_0)^2,
\end{aligned}
\tag{C.5}
$$

where the last inequality uses $\tau \le t$. For any $\tau \in [t]$, we define

$$
y = \big(\langle P_1 - P_2, \sqrt{\lambda_1} \cdot \phi_1\rangle_{\mathcal{H}}, \ldots, \langle P_1 - P_2, \sqrt{\lambda_{d_0}} \cdot \phi_{d_0}\rangle_{\mathcal{H}}\big)^{\top},
\tag{C.6}
$$

$$
v_\tau = \Big(\sqrt{\lambda_1} \cdot \int_{\mathcal{S}} V_\tau(s') \cdot \phi_1(s_\tau, a_\tau, s')\, \mathrm{d}s', \ldots, \sqrt{\lambda_{d_0}} \cdot \int_{\mathcal{S}} V_\tau(s') \cdot \phi_{d_0}(s_\tau, a_\tau, s')\, \mathrm{d}s'\Big)^{\top},
\tag{C.7}
$$

$$
\Lambda_\tau = I_{d_0 \times d_0}/(d_0 K^2 R^2) + \sum_{i=1}^{\tau-1} v_i v_i^{\top}.
\tag{C.8}
$$

Then, it holds that

$$
\begin{aligned}
y^{\top} \Lambda_\tau y &= \frac{1}{d_0 K^2 R^2} \cdot \|y\|_2^2 + \sum_{i=1}^{\tau-1} (y^{\top} v_i)^2 \\
&= \frac{1}{d_0 K^2 R^2} \cdot \sum_{j=1}^{d_0} \langle P_1 - P_2, \sqrt{\lambda_j} \cdot \phi_j\rangle_{\mathcal{H}}^2 + \sum_{i=1}^{\tau-1} |\widetilde{z}_{P_1}(x_i) - \widetilde{z}_{P_2}(x_i)|^2 \\
&\le \frac{1}{d_0 K^2 R^2} \cdot 4 d_0 R^2 + 2/K^2 + 8t \cdot \Gamma(d_0)^2 = 6/K^2 + 8t \cdot \Gamma(d_0)^2,
\end{aligned}
\tag{C.9}
$$

where the inequality uses (C.5) and

$$
\langle P_1 - P_2, \sqrt{\lambda_j} \cdot \phi_j\rangle_{\mathcal{H}}^2 \le \|P_1 - P_2\|_{\mathcal{H}}^2 \cdot \|\sqrt{\lambda_j} \cdot \phi_j\|_{\mathcal{H}}^2 \le 4R^2 \cdot 1 = 4R^2.
$$

Here we uses $\|P\|_{\mathcal{H}} \le R$ for any $P \in \mathcal{P}$ and $\|\sqrt{\lambda_i} \cdot \phi_i\|_{\mathcal{H}} = 1$ for any $j \in [d_0]$.

In the sequel, we establish an upper bound of $|\widetilde{z}_{P_1}(x_\tau) - \widetilde{z}_{P_2}(x_\tau)|$. By the definitions of $y$ in (C.6) and $v_\tau$ in (C.7), we can write $\widetilde{z}_{P_1}(x_\tau) - \widetilde{z}_{P_2}(x_\tau) = \langle y, v_\tau\rangle$. By (C.9), $|\widetilde{z}_{P_1}(z_\tau) - \widetilde{z}_{P_2}(z_\tau)|$ is upper bounded by

$$
\max_{y' \in \mathbb{R}^{d_0}} \langle y', v_\tau\rangle \text{ s.t. } (y')^{\top} \Lambda_\tau y' \le 6/K^2 + 8t \cdot \Gamma(d_0)^2.
$$

The maximizer of such a quadratic program is

$$
y' = \sqrt{[6/K^2 + 8t \cdot \Gamma(d_0)^2]/(v_\tau^{\top} \Lambda_\tau^{-1} v_\tau)} \cdot \Lambda_\tau^{-1} v_\tau.
$$

Therefore, we obtain

$$
|\widetilde{z}_{P_1}(x_\tau) - \widetilde{z}_{P_2}(x_\tau)| \le \sqrt{\big(6/K^2 + 8t \cdot \Gamma(d_0)^2\big)(v_\tau^{\top} \Lambda_\tau^{-1} v_\tau)}.
\tag{C.10}
$$

On the other hand, by (C.2)-(C.4), we have

$$
\begin{aligned}
|\widetilde{z}_{P_1}(x_\tau) - \widetilde{z}_{P_2}(x_\tau)| \\
\ge |z_{P_1}(x_\tau) - z_{P_2}(x_\tau)| &- \big(|\widetilde{z}_{P_1}(x_\tau) - z_{P_1}(x_\tau)| + |\widetilde{z}_{P_2}(x_\tau) - z_{P_2}(x_\tau)|\big) \\
\ge |z_{P_1}(x_\tau) - z_{P_2}(x_\tau)| &- 2\Gamma(d_0) \\
> 1/K - 2\Gamma(d_0). &
\end{aligned}
\tag{C.11}
$$

We set $d_0 = \lceil \widetilde{C} \cdot \log(1/\gamma)/\gamma \cdot \log^{1/\gamma}(4tRKH) \rceil$, where $\widetilde{C}$ is defined in Lemma F.7. Then, by Lemma F.7, it holds that $d_0^\gamma \geq 4(1-\gamma)(\gamma C_2)^{-1}$ and

$$\Gamma(d_0) \leq 1/(4tK). \tag{C.12}$$

Combining (C.10), (C.11), and (C.12), we obtain

$$v_\tau^\top \Lambda_\tau^{-1} v_\tau > \frac{\left(1/K - 2\Gamma(d_0)\right)^2}{6/K^2 + 8t \cdot \Gamma(d_0)^2} \geq \frac{\left(1/K - 1/(2K)\right)^2}{6/K^2 + 1/(2K^2)} > 1/100. \tag{C.13}$$

Therefore, by (C.13), we have

$$
\begin{aligned}
t/100 = \sum_{\tau=1}^t \min\{1/100, v_\tau^\top \Lambda_\tau^{-1} v_\tau\} \\
\leq \sum_{\tau=1}^t 2\log(1 + v_\tau^\top \Lambda_\tau^{-1} v_\tau) \leq 2\big((\log\det(\Lambda_{t+1}) - \log\det(\Lambda_1)\big),
\end{aligned} \tag{C.14}
$$

where the last inequality uses the elliptical potential lemma (Abbasi-Yadkori et al., 2011). Here, similar to (C.8), $\Lambda_{t+1}$ is defined by

$$\Lambda_{t+1} = I_{d_0 \times d_0}/(d_0 K^2 R^2) + \sum_{i=1}^t v_i v_i^\top.$$

Note that

$$\|v_\tau\|_2^2 = \sum_{j=1}^{d_0} \lambda_j \cdot \left(\int_{\mathcal{S}} V_\tau(s')\phi_j(s_\tau, a_\tau, s') \, \mathrm{d}s'\right)^2 \leq d_0 H^2$$

for any $\tau \in [t]$. Thus, setting $\lambda = 1/(d_0 K^2 R^2)$, we have

$$\log\det(\Lambda_1) = d_0 \cdot \log\lambda, \tag{C.15}$$

$$\log\det(\Lambda_{t+1}) \leq d_0 \cdot \log\|\Lambda_{t+1}\|_2 \leq d_0 \cdot \log\left(\lambda + \sum_{i=1}^t \|v_i\|_2^2\right) \leq d_0 \cdot \log(\lambda + d_0 KH^2). \tag{C.16}$$

where the last inequality follows from $t = K \wedge \dim_{\mathrm{E}}(\mathcal{Z}_\mathcal{P}, 1/K)$. Combining (C.15) and (C.16), we have

$$2\big((\log\det(\Lambda_{t+1}) - \log\det(\Lambda_1)\big) \leq 2d_0 \cdot \log(1 + d_0 KH^2/\lambda) = 2d_0 \cdot \log(1 + d_0^2 K^3 R^2 H^2) \tag{C.17}$$

Combining (C.14) and (C.17), we obtain

$$t \leq 200 d_0 \cdot \log(1 + d_0^2 K^3 R^2 H^2) \leq 600 d_0 \cdot \log(d_0 KRH). \tag{C.18}$$

Moreover, by $d_0 = \lceil \widetilde{C} \cdot \log(1/\gamma)/\gamma \cdot \log^{1/\gamma}(4tRKH) \rceil$ and (C.18), there exists an absolute constant $C_8 > 0$ such that

$$
\begin{aligned}
t &\leq 600 d_0 \cdot \log(d_0 KRH) \\
&\leq 600 \cdot \lceil \widetilde{C} \cdot \log(1/\gamma)/\gamma \cdot \log^{1/\gamma}(tKRH)) \rceil \cdot \log\big(\lceil \widetilde{C} \cdot \log(1/\gamma)/\gamma \cdot \log^{1/\gamma}(tKRH) \rceil \cdot KRH\big) \\
&\leq C_8 \cdot \log^2(1/\gamma)/\gamma \cdot \log^{1+1/\gamma}(tKRH). \tag{C.19}
\end{aligned}
$$

Recall that $t = K \wedge \dim_{\mathrm{E}}(\mathcal{Z}_\mathcal{P}, 1/K)$. Thus, by (C.19), we obtain

$$t \leq C_7 \cdot \log^2(1/\gamma)/\gamma \cdot \log^{1+1/\gamma}(KRH) \tag{C.20}$$

for an absolute constant $C_7 > 0$. Thus, we conclude the proof of Lemma C.1. $\qquad\square$

**Lemma C.2.** Under Assumption 4.2, there exists an absolute constant $C_9 > 0$ such that

$$\mathcal{N}_{1/(KH)}(\mathcal{P}, \|\cdot\|_{\infty,1}) \leq C_9 \cdot \log^2(1/\gamma)/\gamma \cdot \log^{1+1/\gamma}(RT)$$

for any $R \geq 2$.

*Proof.* Recall that we define $\Gamma(d_0)$ in (C.4). By choosing $t = 2KH$ in Lemma F.7, there exists

$$d_0 = \lceil \widetilde{C} \cdot \log(1/\gamma)/\gamma \cdot \log^{1/\gamma}(2KRH^2) \rceil$$

where $\widetilde{C}$ is defined in Lemma F.7, such that $d_0^\gamma \geq 4(1 - \gamma)(\gamma C_2)^{-1}$ and

$$\Gamma(d_0) \leq 1/(2KH). \tag{C.21}$$

Here $\gamma$ and $C_2$ are defined in Assumption 4.2. Let $\mathcal{C}$ be a minimal $1/(2d_0^{1/2}KH)$-covering of the Euclidean ball $\{v \in \mathbb{R}^{d_0} : \|v\|_2 \leq R\}$ with respect to the $\ell_2$-norm distance. For any $P \in \mathcal{P}$, we define $\widetilde{P} \in \mathcal{P}$ by

$$\widetilde{P}(x) = \sum_{j=1}^{d_0} \lambda_j \cdot \langle P, \phi_j \rangle_{\mathcal{H}} \cdot \phi_j(x), \text{ for any } x \in \mathcal{Y}.$$

Also, we write

$$v = \big( \sqrt{\lambda_1} \cdot \langle P, \phi_1 \rangle_{\mathcal{H}}, \ldots, \sqrt{\lambda_{d_0}} \cdot \langle P, \phi_{d_0} \rangle_{\mathcal{H}} \big).$$

Since $\{\sqrt{\lambda_j} \cdot \phi_j\}_{j=1}^\infty$ is an orthonormal basis of $\mathcal{H}$ and $P \in \mathcal{H}_R$, we have

$$\|v\|_2 = \|\widetilde{P}\|_{\mathcal{H}} \leq \|P\|_{\mathcal{H}} \leq R.$$

Thus, by the definition of $\mathcal{C}$, there exists $v^* \in \mathcal{C}$ with $\|v - v^*\|_2 \leq 1/(2d_0^{1/2}KH)$. We define $P^* \in \mathcal{P}$ by

$$P^*(x) = \sum_{j=1}^{d_0} v_j^* \cdot \sqrt{\lambda_j} \cdot \phi_j(x), \text{ for any } x \in \mathcal{Y}.$$

Then, by Assumption 4.2, for any $x \in \mathcal{Y}$, we have

$$|\widetilde{P}(x) - P^*(x)| = \Big| \sum_{j=1}^{d_0} (v_j - v_j^*) \cdot \sqrt{\lambda_j} \cdot \phi_j(x) \Big|$$

$$\leq \|v - v^*\|_1 \leq d_0^{1/2} \cdot \|v - v^*\|_2 \leq 1/(2KH). \tag{C.22}$$

Also, for any $x \in \mathcal{Y}$, we have

$$|P(x) - \widetilde{P}(x)| = \Big| \sum_{j > d_0} \sqrt{\lambda_j} \cdot \phi_j(x) \cdot \langle P, \sqrt{\lambda_j} \cdot \phi_j \rangle_{\mathcal{H}} \Big|$$

$$\leq \sum_{j > d_0} \sqrt{\lambda_j} \cdot R$$

$$\leq \sum_{j > d_0} \sqrt{C_1} \cdot \exp(-C_2 j^\gamma/2) \cdot R \leq \Gamma(d_0), \tag{C.23}$$

where the first inequality uses $\|P\|_{\mathcal{H}} \leq R$ and $\|\sqrt{\lambda_j} \cdot \phi_j\|_{\mathcal{H}} = 1$, the second inequality follows from Assumption 4.2, and the last inequality follows from the same argument in (C.3) and $H \geq 1$. Thus,

for any $x \in \mathcal{Y}$, it holds that

$$|P(x) - P^*(x)| \leq |P(x) - \widetilde{P}(x)| + |\widetilde{P}(x) - P^*(x)| \leq \Gamma(d_0) + 1/(2KH) \leq 1/(KH). \quad \text{(C.24)}$$

We define

$$\mathcal{P}_\mathcal{C} = \left\{ P : \exists \, v \in \mathcal{C} \text{ s.t. } P(x) = \sum_{j=1}^{d_0} v_j \cdot \sqrt{\lambda_j} \cdot \phi_j(x) \text{ for any } x \in \mathcal{Y} \right\}.$$

Then, by (C.24), $\mathcal{P}_\mathcal{C}$ is a $1/(KH)$-covering of $\mathcal{P}$ with respect to the $\ell_\infty$-norm distance. Therefore, we have

$$\mathcal{N}_{1/(KH)}(\mathcal{P}, \|\cdot\|_{\infty,1}) \leq \mathcal{N}_{1/(KH)}(\mathcal{P}, \|\cdot\|_\infty)$$
$$\leq |\mathcal{P}_\mathcal{C}| \leq |\mathcal{C}| \leq C_{12} \cdot d_0 \cdot \log(d_0 KRH) \leq C_9 \cdot \log^2(1/\gamma)/\gamma \cdot \log^{1+1/\gamma}(KRH), \quad \text{(C.25)}$$

where $C_{12}, C_9 > 0$ are absolute constants. Here the first inequality is because $\|\cdot\|_{\infty,1} \leq \|\cdot\|_\infty$ given $|\mathcal{S}| \leq 1$, the third inequality follows from Corollary 4.2.13 of Vershynin (2018), and the last inequality follows from the same argument in (C.19). Thus, we conclude the proof of Lemma C.2. $\qquad\square$

We note that our proofs of Lemmas C.1 and C.2 for the exponential decay case can be made general. For a finite-rank kernel, we can let $d_0$ be the rank of the kernel, i.e., the number of nonzero eigenvalues. Then, the same proof still holds and we can show that the eluder dimension is upper bounded by $\mathcal{O}(d_0 \log(RKH))$. Moreover, for a kernel with polynomially decaying eigenvalues, that is, we have

$$\lambda_j \leq C_{\text{poly}} \cdot j^{-\gamma} \text{ for some constants } C_{\text{poly}}, \gamma > 0 \text{ and any } j \geq 1,$$

we can still truncate at dimension $d_0$ and calculate $\Gamma(d_0)$ in (C.4) using the polynomial eigenvalue decay condition. It can be shown that $\Gamma(d_0)$ decays polynomially in $d_0$. Then, we can find a proper $d_0$ by solving (C.12) and (C.21) and follow the same proof afterwards.

**Proof of Theorem 4.3**

*Proof.* Recall that $d = K \wedge \dim_{\mathrm{E}}(\mathcal{Z}_\mathcal{P}, 1/K)$. By Theorem 4.12, it holds that

$$\text{Regret}(T) \leq \sqrt{2H^3 T \cdot \log |\mathcal{A}|} + \sqrt{32 H^2 T \cdot \log(p/2)} + H(dH + 1) + 4\sqrt{d\beta HT}, \quad \text{(C.26)}$$

with probability at least $(1 - p)$, where we set $\alpha = \sqrt{2 \log |\mathcal{A}| / (KH^2)}$ and

$$\beta \geq 2H^2 \cdot \log\left( \mathcal{N}_{1/T}(\mathcal{P}, \|\cdot\|_{\infty,1}) \cdot 2H/p \right) + 4\left( H + \sqrt{H^2/4 \cdot \log(8K^2 H/p)} \right). \quad \text{(C.27)}$$

By Lemma C.2, we have

$$\mathcal{N}_{1/(KH)}(\mathcal{P}, \|\cdot\|_{\infty,1}) \leq C_9 \cdot \log^2(1/\gamma)/\gamma \cdot \log^{1+1/\gamma}(RT),$$

which implies there exists an absolute constant $C_3 > 0$ such that

$$2H^2 \cdot \log\left( \mathcal{N}_{1/(KH)}(\mathcal{P}, \|\cdot\|_{\infty,1}) \cdot 2H/p \right) + 4\left( H + \sqrt{H^2/4 \cdot \log(8K^2 H/p)} \right)$$
$$\leq C_3 H^2 \cdot \log^2(1/\gamma)/\gamma \cdot \log^{1+1/\gamma}(RT/p).$$

In other words, (C.27) holds if we set

$$\beta = C_3 H^2 \cdot \log^2(1/\gamma)/\gamma \cdot \log^{1+1/\gamma}(RT/p). \quad \text{(C.28)}$$

On the other hand, by Lemma C.1, we have

$$d \leq C_7 \cdot \log^2(1/\gamma)/\gamma \cdot \log^{1+1/\gamma}(RT). \quad \text{(C.29)}$$

Plugging (C.28) and (C.29) into (C.26), we obtain

$$
\begin{aligned}
\text{Regret}(T) &\leq \sqrt{2H^3 T \cdot \log|\mathcal{A}|} + \sqrt{32H^2 T \cdot \log(p/2)} + H \\
&\quad + C_7 H^2 \cdot \log^2(1/\gamma)/\gamma \cdot \log^{1+1/\gamma}(RT) \\
&\quad + 4\sqrt{C_3 C_7 H^3 T} \cdot \log^2(1/\gamma)/\gamma \cdot \log^{(1+1/\gamma)/2}(RT) \cdot \log^{(1+1/\gamma)/2}(RT/p) \\
&\leq C_4 \sqrt{H^3 T} \cdot \log^2(1/\gamma)/\gamma \cdot \log^{1+1/\gamma}(|\mathcal{A}|RT/p) \qquad\qquad\text{(C.30)}
\end{aligned}
$$

with probability at least $(1-p)$, where $C_4 > 0$ is an absolute constant. Thus, we conclude the proof of Theorem 4.3. $\qquad\square$

## D    PROOF OF THEOREM 4.7

In this section, we prove Theorem 4.7. Similar to the proof of Theorem 4.3, we upper bound the eluder dimension of $\mathcal{Z}_\mathcal{P}$ and log-covering number of $\mathcal{P}$ by the following two lemmas respectively.

Recall that $\mathcal{B}_R$ is defined in (4.1). By the definition of $\mathcal{K}_{\text{NTK}}$ in (4.2), we have

$$
\mathcal{H}_R = \{P \,:\, \exists\, w \in \mathcal{B}_R \text{ s.t. } P(x) = \nabla_w \text{NN}(x; w^0)^\top (w - w^0) \text{ for any } x \in \mathcal{Y}\}.
$$

Without loss of generality, we assume entries of $\nabla_w \text{NN}(\cdot; w^0)$ are linearly independent, which happens with probability one for most neural networks with nonlinear activation functions and random initialization.

**Lemma D.1.** Under Assumptions 4.6 and 4.4, there exists an absolute constant $C_{10}$ such that

$$
K \wedge \dim_E(\mathcal{Z}_\mathcal{P}, 1/K) \leq C_{10} \cdot \log^2(1/\gamma)/\gamma \cdot \log^{1+1/\gamma}(RT).
$$

*Proof.* Let $t = K \wedge \dim_E(\mathcal{Z}_\mathcal{P}, 1/K)$. By Definition 4.8, there exists a sequence $x_1, \ldots, x_t \in \mathcal{S} \times \mathcal{A} \times [0, H]^\mathcal{S}$ such that $x_\tau$ is $(\mathcal{Z}_\mathcal{P}, 1/K)$-independent of $x_1, \ldots, x_{\tau-1}$ for any $\tau \in [t]$. In other words, for any $\tau \in [t]$, there exist $P_1, P_2 \in \mathcal{P}$ such that

$$
\sum_{i=1}^{\tau-1} |z_{P_1}(x_i) - z_{P_2}(x_i)|^2 \leq 1/K^2, \qquad\qquad\text{(D.1)}
$$

$$
|z_{P_1}(x_\tau) - z_{P_2}(x_\tau)| > 1/K. \qquad\qquad\text{(D.2)}
$$

For any $x = (s, a, V) \in \mathcal{S} \times \mathcal{A} \times [0, H]^\mathcal{S}$ and $P \in \mathcal{P}$, we define

$$
\bar{P}(x; w) = \nabla_w \text{NN}(x; w^0)^\top (w - w^0).
$$

Using the reproducing property of $\mathcal{H}$ and the representation of $\mathcal{K}$ in (2.2), we have

$$
\begin{aligned}
z_P(x) &= \int_\mathcal{S} V(s') \cdot P(s, a, s'; w)\,\mathrm{d}s' \\
&= \int_\mathcal{S} V(s') \cdot \bar{P}(s, a, s'; w)\,\mathrm{d}s' + \int_\mathcal{S} V(s') \cdot \big(P(s, a, s'; w) - \bar{P}(s, a, s'; w)\big)\,\mathrm{d}s' \\
&= \sum_{j=1}^\infty \sqrt{\lambda_j} \cdot \int_\mathcal{S} V(s') \cdot \phi_j(s, a, s')\,\mathrm{d}s' \cdot \langle \bar{P}, \sqrt{\lambda_j} \cdot \phi_j \rangle_\mathcal{H} \\
&\quad + \int_\mathcal{S} V(s') \cdot \big(P(s, a, s'; w) - \bar{P}(s, a, s'; w)\big)\,\mathrm{d}s'.
\end{aligned}
$$

For any $d_0$ such that $d_0^\gamma \geq 4(1-\gamma)(\gamma C_2)^{-1}$, we define

$$
\widetilde{z}_P(x) = \sum_{1 \leq j \leq d_0} \sqrt{\lambda_j} \cdot \int_\mathcal{S} V(s') \cdot \phi_j(s, a, s')\,\mathrm{d}s' \cdot \langle \bar{P}, \sqrt{\lambda_j} \cdot \phi_j \rangle_\mathcal{H}
$$

for any $x = (s, a, V) \in \mathcal{S} \times \mathcal{A} \times [0, H]^{\mathcal{S}}$ and $P \in \mathcal{P}$. And we have

$$
\begin{aligned}
|z_P(x) - \widetilde{z}_P(x)| &\leq \Big| \sum_{j > d_0} \sqrt{\lambda_j} \cdot \int_{\mathcal{S}} V(s') \cdot \phi_j(s, a, s') \, ds' \cdot \langle \bar{P}, \sqrt{\lambda_j} \cdot \phi_j \rangle_{\mathcal{H}} \Big| \\
&\quad + \Big| \int_{\mathcal{S}} V(s') \big( P(s, a, s'; w) - \bar{P}(s, a, s'; w) \big) \, ds' \Big| \\
&\leq \sum_{j > d_0} \sqrt{\lambda_j} \cdot H \cdot R + \xi_m H \\
&\leq 4 d_0^{1-\gamma} \sqrt{C_1} H R (\gamma C_2)^{-1} \cdot \exp(-C_2 d_0^\gamma / 2) + \xi_m H = \Gamma(d_0) + \xi_m H, \quad \text{(D.3)}
\end{aligned}
$$

where the second inequality uses Assumption 4.2, $\|\bar{P}\|_{\mathcal{H}} \leq R$, $\|\sqrt{\lambda_j} \cdot \phi_j\|_{\mathcal{H}} = 1$, and the definition of $\xi_m$ in Condition 4.4, the third inequality follows from Lemma F.6, and $\Gamma(d_0)$ is defined in (C.4). Then, using the triangle inequality, we have

$$
\begin{aligned}
\sum_{i=1}^{\tau-1} |\widetilde{z}_{P_1}(x_i) - \widetilde{z}_{P_2}(x_i)|^2 &\leq \sum_{i=1}^{\tau-1} \big( |z_{P_1}(x_i) - z_{P_2}(x_i)| + 2\Gamma(d_0) + 2\xi_m H \big)^2 \\
&\leq \sum_{i=1}^{\tau-1} 2 |z_{P_1}(x_i) - z_{P_2}(x_i)|^2 + 16t \cdot \Gamma(d_0)^2 + 16t \xi_m^2 H^2 \\
&\leq 2/K^2 + 16t \cdot \Gamma(d_0)^2 + 16t \xi_m^2 H^2, \quad \text{(D.4)}
\end{aligned}
$$

where the last inequality is by (D.1).

In the sequel, we establish an upper bound of $|\widetilde{z}_{P_1}(x_\tau) - \widetilde{z}_{P_2}(x_\tau)|$. For any $\tau \in [t]$, similar to (C.6)-(C.8), we denote

$$
y = \big( \langle P_1 - P_2, \sqrt{\lambda_1} \cdot \phi_1 \rangle_{\mathcal{H}}, \ldots, \langle \overline{P}_1 - \overline{P}_2, \sqrt{\lambda_{d_0}} \cdot \phi_{d_0} \rangle_{\mathcal{H}} \big)^\top, \quad \text{(D.5)}
$$

$$
v_\tau = \Big( \sqrt{\lambda_1} \cdot \int_{\mathcal{S}} V_\tau(s') \cdot \phi_1(s_\tau, a_\tau, s') \, ds', \ldots, \sqrt{\lambda_{d_0}} \cdot \int_{\mathcal{S}} V_\tau(s') \cdot \phi_{d_0}(s_\tau, a_\tau, s') \, ds' \Big)^\top, \quad \text{(D.6)}
$$

$$
\Lambda_\tau = I_{d_0 \times d_0} / (d_0 K^2 R^2) + \sum_{i=1}^{\tau-1} v_i v_i^\top. \quad \text{(D.7)}
$$

Then, we have

$$
\begin{aligned}
y^\top \Lambda_\tau y &= \frac{1}{d_0 K^2 R^2} \cdot \|y\|_2^2 + \sum_{i=1}^{\tau-1} (y^\top v_i)^2 \\
&= \frac{1}{d_0 K^2 R^2} \cdot \sum_{j=1}^{d_0} \langle \overline{P}_1 - \overline{P}_2, \sqrt{\lambda_j} \cdot \phi_j \rangle_{\mathcal{H}}^2 + \sum_{i=1}^{\tau-1} |\widetilde{z}_{P_1}(x_i) - \widetilde{z}_{P_2}(x_i)|^2 \\
&\leq \frac{1}{d_0 K^2 R^2} \cdot 4 d_0 R^2 + 2/K^2 + 16t \cdot \Gamma(d_0)^2 + 16t e_m^2 H^2 \\
&= 6/K^2 + 16t \cdot \Gamma(d_0)^2 + 16t e_m^2 H^2 \leq 7/K^2 + 16t \cdot \Gamma(d_0)^2. \quad \text{(D.8)}
\end{aligned}
$$

Here the inequality uses (D.4), Condition 4.4, and

$$
\langle \overline{P}_1 - \overline{P}_2, \sqrt{\lambda_j} \cdot \phi_j \rangle_{\mathcal{H}}^2 \leq \|\overline{P}_1 - \overline{P}_2\|_{\mathcal{H}}^2 \cdot \|\sqrt{\lambda_j} \cdot \phi_j\|_{\mathcal{H}}^2 \leq 4R^2 \cdot 1 = 4R^2,
$$

which follows from the fact $\|\overline{P}_1\|_{\mathcal{H}}, \|\overline{P}_2\|_{\mathcal{H}} \leq R$ and $\|\sqrt{\lambda_i} \cdot \phi_i\|_{\mathcal{H}} = 1$ for any $j \in [d_0]$. By the definitions of $y$ in (D.5) and $v_\tau$ in (D.6), we can write $\widetilde{z}_{P_1}(x_\tau) - \widetilde{z}_{P_2}(x_\tau) = \langle y, v_\tau \rangle$, which by (D.8) is upper bounded by

$$
\max_{y' \in \mathbb{R}^{d_0}} \langle y', v_\tau \rangle \text{ s.t. } (y')^\top \Lambda_\tau y' \leq 7/K^2 + 16t \cdot \Gamma(d_0)^2.
$$

The maximizer of such a quadratic program is

$$y' = \sqrt{\left(7/K^2 + 16t \cdot \Gamma(d_0)^2\right)/(v_\tau^\top \Lambda_\tau^{-1} v_\tau)} \cdot \Lambda_\tau^{-1} v_\tau.$$

Therefore, we have

$$|\widetilde{z}_{P_1}(x_\tau) - \widetilde{z}_{P_2}(x_\tau)| \leq \sqrt{\left(7/K^2 + 16t \cdot \Gamma(d_0)^2\right)(v_\tau^\top \Lambda_\tau^{-1} v_\tau)}. \tag{D.9}$$

On the other hand, by (D.2), (D.3), and Condition 4.4 we have

$$\begin{aligned}
|\widetilde{z}_{P_1}(x_\tau) - \widetilde{z}_{P_2}(x_\tau)| \\
&\geq |z_{P_1}(x_\tau) - z_{P_2}(x_\tau)| - \left(|\widetilde{z}_{P_1}(x_\tau) - z_{P_2}(x_\tau)| + |\widetilde{z}_{P_1}(x_\tau) - z_{P_2}(x_\tau)|\right) \\
&\geq |z_{P_1}(x_\tau) - z_{P_2}(x_\tau)| - 2\Gamma(d_0) - 2\xi_m H \\
&> 1/K - 2\Gamma(d_0) - 1/(2K) \geq 1/(2K) - 2\Gamma(d_0). \tag{D.10}
\end{aligned}$$

We set $d_0 = \lceil \widetilde{C} \cdot \log(1/\gamma)/\gamma \cdot \log^{1/\gamma}(8tKRH) \rceil$, where $\widetilde{C}$ is defined in Lemma F.7. Then, by Lemma F.7 it holds that $d_0^\gamma \geq 4(1-\gamma)(\gamma C_2)^{-1}$ and

$$\Gamma(d_0) \leq 1/(8tK). \tag{D.11}$$

Combining (D.9)-(D.11), we obtain

$$v_\tau^\top \Lambda_\tau^{-1} v_\tau > \frac{\left(1/(2K) - 2\Gamma(d_0)\right)^2}{7/K^2 + 16t \cdot \Gamma(d_0)^2} \geq \frac{\left(1/(2K) - 1/(4K)\right)^2}{7/K^2 + 1/(4K^2)} > 1/100.$$

Following the same argument in (C.14)-(C.20) in the proof of Lemma C.1, we obtain

$$t \leq C_{10} \cdot \log^2(1/\gamma)/\gamma \cdot \log^{1+1/\gamma}(KRH)$$

for an absolute constant $C_{10} > 0$. Thus, we conclude the proof of Lemma D.1. $\qquad\square$

**Lemma D.2.** Under Assumption 4.6 and 4.4, there exists an absolute constant $C_{11}$ such that we have

$$\mathcal{N}_{1/(KH)}(\mathcal{P}, \|\cdot\|_{\infty,1}) \leq C_{11} \cdot \log^2(1/\gamma)/\gamma \cdot \log^{1+1/\gamma}(RT)$$

for any $R \geq 2$.

*Proof.* By choosing $t = 4KH$ in Lemma F.7, there exists $d_0 = \lceil \widetilde{C} \cdot \log(1/\gamma)/\gamma \cdot \log^{1/\gamma}(4KRH^2) \rceil$, where $\widetilde{C}$ is defined in Lemma F.7, such that $d_0^\gamma \geq 4(1-\gamma)(\gamma C_2)^{-1}$ and

$$\Gamma(d_0) \leq 1/(4KH). \tag{D.12}$$

Here $\Gamma(d_0)$ is defined in (C.4), $\gamma$ and $C_2$ are defined in Assumption 4.6. Let $\mathcal{C}$ be a minimal $1/(4d_0^{1/2}KH)$-covering of the Euclidean ball $\{v \in \mathbb{R}^{d_0} : \|v\|_2 \leq R\}$ with respect to the $\ell_2$-norm distance. For any $P \in \mathcal{P}$ with parameter $w$, we define $\bar{P}$ and $\widetilde{P}$ by

$$\bar{P}(x) = \nabla_w \mathrm{NN}(x; w^0)^\top (w - w^0),$$

$$\widetilde{P}(x) = \sum_{j=1}^{d_0} \lambda_j \cdot \langle \bar{P}, \phi_j \rangle_{\mathcal{H}} \cdot \phi_j(x).$$

for any $x \in \mathcal{Y}$. Also, we write

$$v = \left(\sqrt{\lambda_1} \cdot \langle \bar{P}, \phi_1 \rangle_{\mathcal{H}}, \ldots, \sqrt{\lambda_{d_0}} \cdot \langle \bar{P}, \phi_{d_0} \rangle_{\mathcal{H}}\right).$$

Note that, because $\{\sqrt{\lambda_j} \cdot \phi_j\}_{j=1}^{\infty}$ is an orthonormal basis in $\mathcal{H}$ and $w \in \mathcal{B}_R$, we have

$$\|v\|_2 = \|\widetilde{P}\|_{\mathcal{H}} \leq \|\bar{P}\|_{\mathcal{H}} \leq R.$$

Thus, there exists $v^* \in \mathcal{C}$ with $\|v - v^*\|_2 \leq 1/(4d_0^{1/2}KH)$. We define $P^* \in \mathcal{H}_R$ by

$$P^*(x) = \sum_{j=1}^{d_0} v_j^* \cdot \sqrt{\lambda_j} \cdot \phi_j(x).$$

By the same arguments in (C.22) and (C.23), for any $x \in \mathcal{Y}$, we have

$$|\bar{P}(x) - \widetilde{P}(x)| \leq \Gamma(d_0), \quad |\widetilde{P}(x) - P^*(x)| \leq 1/(4KH). \tag{D.13}$$

Then, by (D.12), (D.13), and Condition 4.4, for any $x \in \mathcal{Y}$, it holds that

$$
\begin{aligned}
|P(x) - P^*(x)| &\leq |P(x) - \bar{P}(x)| + |\bar{P}(x) - \widetilde{P}(x)| + |\widetilde{P}(x) - P^*(x)| \\
&\leq \xi_m + \Gamma(d_0) + 1/(4KH) \leq 3/(4KH).
\end{aligned}
\tag{D.14}
$$

Moreover, since $P^* \in \mathcal{H}_R$, there exists a $w_v \in \mathcal{B}_R$ such that

$$P^*(x) = \nabla_w \mathrm{NN}(x; w^0)^\top (w_v - w^0)$$

for any $x \in \mathcal{Y}$. By Condition 4.4, we have

$$|P^*(x) - \mathrm{NN}(x; w_v)| \leq \xi_m \leq 1/(4KH). \tag{D.15}$$

Combining (D.14) and (D.15), we have

$$|P(x) - \mathrm{NN}(x; w_v)| \leq 1/(KH). \tag{D.16}$$

Note that because entries of $\nabla_w \mathrm{NN}(\cdot; w^0)$ are linearly independent, $w_v$ is unique for any $v : \|v\|_2 \leq R$. We define

$$\mathcal{P}_{\mathcal{C}} = \Big\{ P : \exists v \in \mathcal{C} \text{ s.t. } P(x) = \mathrm{NN}(x; w_v) \text{ for any } x \in \mathcal{Y} \Big\}.$$

Then, by (D.16), $\mathcal{P}_{\mathcal{C}}$ is a $1/(KH)$-covering of $\mathcal{P}$ with respect to the $\ell_\infty$-norm distance. Following the same argument of (C.25), we have

$$
\begin{aligned}
\mathcal{N}_{1/(KH)}(\mathcal{P}, \|\cdot\|_{\infty,1}) &\leq \mathcal{N}_{1/(KH)}(\mathcal{P}, \|\cdot\|_\infty) \\
&\leq |\mathcal{P}_{\mathcal{C}}| \leq |\mathcal{C}| \leq C_{13} \cdot d_0 \cdot \log(d_0 KRH) \leq C_{11} \cdot \log^2(1/\gamma)/\gamma \cdot \log^{1+1/\gamma}(KRH),
\end{aligned}
$$

where $C_{13}, C_{11} > 0$ are absolute constants. Thus, we conclude the proof of Lemma D.2. $\qquad \square$

**Proof of Theorem 4.7**

*Proof.* Recall that $d = K \wedge \dim_{\mathrm{E}}(\mathcal{Z}_{\mathcal{P}}, 1/K)$. By the result of a general $\mathcal{P}$ in Theorem 4.12, we have

$$\mathrm{Regret}(T) \leq \sqrt{2H^3 T \cdot \log |\mathcal{A}|} + \sqrt{32H^2 T \cdot \log(p/2)} + H(dH + 1) + 4\sqrt{d\beta HT}, \tag{D.17}$$

with probability at least $(1 - p)$, when we set $\alpha = \sqrt{2 \log |\mathcal{A}|/(KH^2)}$ and

$$\beta \geq 2H^2 \cdot \log\big(\mathcal{N}_{1/T}(\mathcal{P}, \|\cdot\|_{\infty,1}) \cdot 2H/p\big) + 4\big(H + \sqrt{H^2/4 \cdot \log(8K^2 H/p)}\big) \tag{D.18}$$

By Lemma D.2, we have

$$\mathcal{N}_{1/(KH)}(\mathcal{P}, \|\cdot\|_{\infty,1}) \leq C_{11} \cdot \log^2(1/\gamma)/\gamma \cdot \log^{1+1/\gamma}(RT),$$

which implies that there exists an absolute constant $C_5$ such that

$$2H^2 \cdot \log\big(\mathcal{N}_{1/(KH)}(\mathcal{P}, \|\cdot\|_{\infty,1}) \cdot 2H/p\big) + 4\big(H + \sqrt{H^2/4 \cdot \log(8K^2H/p)}\big)$$
$$\leq C_5 H^2 \cdot \log^2(1/\gamma)/\gamma \cdot \log^{1+1/\gamma}(RT/p).$$

In other words, (D.18) holds if we set

$$\beta \geq C_5 H^2 \cdot \log^2(1/\gamma)/\gamma \cdot \log^{1+1/\gamma}(RT/p). \tag{D.19}$$

On the other hand, by Lemma D.1, we have

$$d \leq C_{10} \cdot \log^2(1/\gamma)/\gamma \cdot \log^{1+1/\gamma}(RT). \tag{D.20}$$

Plugging (D.19) and (D.20) into (D.17), we obtain

$$\begin{aligned}
\text{Regret}(T) \leq{}& \sqrt{2H^3 T \log|\mathcal{A}|} + \sqrt{32H^2 T \log(p/2)} + H \\
&+ C_{10} H^2 \cdot \log^2(1/\gamma)/\gamma \cdot \log^{1+1/\gamma}(RT) \\
&+ 4\sqrt{C_5 C_{10} H^3 T} \cdot \log^2(1/\gamma)/\gamma \cdot \log^{(1+1/\gamma)/2}(RT) \cdot \log^{(1+1/\gamma)/2}(RT/p) \\
\leq{}& C_6 \sqrt{H^3 T} \cdot \log^2(1/\gamma)/\gamma \cdot \log^{1+1/\gamma}(|\mathcal{A}|RT/p)
\end{aligned}$$

with probability at least $(1-p)$, where $C_6 > 0$ is an absolute constant. Thus, we conclude the proof of Theorem 4.7. $\qquad\square$

## E  IMPLICIT LINEARIZATION

**Two-layer fully-connected neural networks:**  A two-layer fully-connected neural network is defined by

$$\text{NN}(x; w) = \frac{1}{\sqrt{m/d_{\mathcal{Y}}}} \sum_{j=1}^{m/d_{\mathcal{Y}}} b_j \cdot \sigma(x^\top w_j), \tag{E.1}$$

where, without loss of generality, we assume $m$ is integer times of $2d_{\mathcal{Y}}$. Here $\sigma$ is the activation function. The weight vectors $w$ and $b$ corresponding to the first layer and second layer, respectively, take the form

$$w = (w_1^\top, \ldots, w_{m/d_{\mathcal{Y}}}^\top)^\top \in \mathbb{R}^m, \quad b = (b_1, \ldots, b_{m/d_{\mathcal{Y}}})^\top \in \mathbb{R}^{m/d_{\mathcal{Y}}},$$

respectively. During the training, we only tune the weights in $w$.

**Symmetric Initialization:**  When initializing the neural network, we generate the initial weight vectors $w^0$ and $b$ by

$$w_j^0 \overset{\text{i.i.d.}}{\sim} \mathcal{N}(0, 1/d_{\mathcal{Y}} \cdot I_{d_{\mathcal{Y}} \times d_{\mathcal{Y}}}), \quad w_{j+m/(2d_{\mathcal{Y}})}^0 = w_j^0,$$
$$b_j \overset{\text{i.i.d.}}{\sim} \text{Unif}(\{-1, 1\}), \quad b_{j+m/(2d_{\mathcal{Y}})} = -b_j$$

for any $j \in [m/(2d_{\mathcal{Y}})]$. As a result of such initialization, we have $\text{NN}(\cdot; w^0) = 0$ and we can generalize the result to multilayer neural networks by setting the last two layers in this manner.

**Proof of Lemma 4.5**

*Proof.* Let NN be a two-layer fully-connected neural network in the form (E.1). The activation function $\sigma$ is 1-smooth and the second layer weights satisfies $b_j \in \{-1, 1\}$ for any $j \in [m/d_{\mathcal{Y}}]$.

For any $w \in \mathbb{R}^m$ such that $\|w - w^0\|_2 \leq R$, we have

$$\|\nabla_w \mathrm{NN}(x; w) - \nabla_w \mathrm{NN}(x; w^0)\|_2^2$$

$$= \frac{1}{m/d_{\mathcal{Y}}} \sum_{j=1}^{m/d_{\mathcal{Y}}} \|b_j \sigma'(x^\top w_j) \cdot x - b_j \sigma'(x^\top w_j^0) \cdot x\|_2^2$$

$$= \frac{1}{m/d_{\mathcal{Y}}} \sum_{j=1}^{m/d_{\mathcal{Y}}} \left(\sigma'(x^\top w_j) - \sigma'(x^\top w_j^0)\right)^2 \cdot (b_j)^2 \cdot \|x\|_2^2$$

Note that, by the assumption that $\sigma$ is 1-smooth, we have

$$\left(\sigma'(x^\top w_j) - \sigma'(x^\top w_j^0)\right)^2 \leq (x^\top w_j - x^\top w_j^0)^2 \leq \|w_j - w_j^0\|_2^2.$$

Thus, we have

$$\|\nabla_w \mathrm{NN}(x; w) - \nabla_w \mathrm{NN}(x; w^0)\|_2^2 \leq \frac{1}{m/d_{\mathcal{Y}}} \sum_{j=1}^{m/d_{\mathcal{Y}}} \|w_j - w_j^0\|_2^2 \leq d_{\mathcal{Y}} R^2 / m. \qquad \text{(E.2)}$$

By the mean value theorem, for any $w : \|w - w^0\|_2 \leq R$, there exists $w^\dagger : \|w^\dagger - w^0\|_2 \leq R$ such that

$$\mathrm{NN}(x; w) - \mathrm{NN}(x; w^0)$$

$$= \nabla_w \mathrm{NN}(x; w^\dagger)^\top (w - w^0)$$

$$= \nabla_w \mathrm{NN}(x; w^0)^\top (w - w^0) + \left(\nabla_w \mathrm{NN}(x; w^\dagger) - \nabla_w \mathrm{NN}(x; w^0)\right)^\top (w - w^0),$$

combining which with (E.2) we obtain

$$|\mathrm{NN}(x; w) - \nabla_w \mathrm{NN}(x; w^0)^\top (w - w^0)|$$

$$= |\mathrm{NN}(x; w) - \mathrm{NN}(x; w^0) - \nabla_w \mathrm{NN}(x; w^0)^\top (w - w^0)|$$

$$= \left|\left(\nabla_w \mathrm{NN}(x; w^\dagger) - \nabla_w \mathrm{NN}(x; w^0)\right)^\top (w - w^0)\right|$$

$$\leq \|\nabla_w \mathrm{NN}(x; w^\dagger) - \nabla_w \mathrm{NN}(x; w^0)\|_2 \cdot \|w - w^0\|_2 \leq d_{\mathcal{Y}}^{1/2} R^2 m^{-1/2}.$$

Therefore, we have $\xi_m \leq d_{\mathcal{Y}}^{1/2} R^2 m^{-1/2}$. Then, Condition 4.4 holds when

$$m \geq d_{\mathcal{Y}} R^4 K^3 H^2.$$

Thus, we conclude the proof of Lemma 4.5. $\qquad \square$

# F SUPPORTING LEMMAS

## F.1 DECOMPOSITION

For notational simplicity, we define the linear operator $\mathbb{J}_h^k$ for $(k, h) \in [K] \times [H]$ by

$$(\mathbb{J}_h^k f)(s) = \mathbb{E}[f(s, a) \,|\, a \sim \pi_h^k(\cdot \,|\, s)], \quad \text{for any } s \in \mathcal{S}, \ f \in [0, H]^{\mathcal{S} \times \mathcal{A}}.$$

**Lemma F.1** (Martingale Decomposition). *For any $k \in [K]$, we have*

$$V_1^k(s_1^k) - V_1^{\pi^k}(s_1^k) = \sum_{h=1}^{H} (D_{h,1}^k + D_{h,2}^k) + \sum_{h=1}^{H} \left(Q_h^k(s_h^k, a_h^k) - \left(r_h^k(s_h^k, a_h^k) + \mathbb{P}_h V_{h+1}^k(s_h^k, a_h^k)\right)\right),$$

where $D_{h,1}^k$ and $D_{h,2}^k$ take the forms

$$D_{h,1}^k = \big(\mathbb{J}_h^k(Q_h^k - Q_h^{\pi^k,k})\big)(s_h^k) - (Q_h^k - Q_h^{\pi^k,k})(s_h^k, a_h^k),$$

$$D_{h,2}^k = (\mathbb{P}_h V_{h+1}^k - \mathbb{P}_h V_{h+1}^{\pi^k,k})(s_h^k, a_h^k) - (V_{h+1}^k - V_{h+1}^{\pi^k,k})(s_{h+1}^k).$$

Moreover, we have $D_{H,2}^k = 0$ for any $k \in [K]$, and the sequence

$$D_{1,1}^1, D_{1,2}^1 + D_{2,1}^1, D_{2,2}^1 + D_{3,1}^1, \ldots, D_{H-1,2}^1 + D_{H,1}^1,$$

$$D_{1,1}^2, D_{1,2}^2 + D_{2,1}^2, D_{2,2}^2 + D_{3,1}^2, \ldots, D_{H-1,2}^2 + D_{H,1}^2,$$

$$\cdots \cdots \tag{F.1}$$

is a martingale difference sequence with respect to the filtration $\{\widetilde{\mathcal{F}}_t\}_{t \geq 1}$ in Definition F.2 and each term is bounded by $4H$.

*Proof.* For any $(k,h) \in [K] \times [H]$, by the definition of the operator $\mathbb{J}_h^k$, we have

$$V_h^k(s_h^k) - V_h^{\pi^k}(s_h^k)$$
$$= (\mathbb{J}_h^k Q_h^k)(s_h^k) - (\mathbb{J}_h^k Q_h^{\pi^k,k})(s_h^k)$$
$$= (Q_h^k - Q_h^{\pi^k,k})(s_h^k, a_h^k) + \underbrace{\big(\mathbb{J}_h^k(Q_h^k - Q_h^{\pi^k,k})\big)(s_h^k) - (Q_h^k - Q_h^{\pi^k,k})(s_h^k, a_h^k)}_{= D_{h,1}^k}, \tag{F.2}$$

where we denote the second term on the right-hand side by $D_{h,1}^k$. Also, we have

$$(Q_h^k - Q_h^{\pi^k,k})(s_h^k, a_h^k)$$
$$= (Q_h^k - r_h^k - \mathbb{P}_h V_{h+1}^{\pi^k,k})(s_h^k, a_h^k)$$
$$= (\mathbb{P}_h V_{h+1}^k - \mathbb{P}_h V_{h+1}^{\pi^k,k})(s_h^k, a_h^k) + (Q_h^k - r_h^k - \mathbb{P}_h V_{h+1}^k)(s_h^k, a_h^k)$$
$$= V_{h+1}^k(s_{h+1}^k) - V_{h+1}^{\pi^k,k}(s_{h+1}^k) + (Q_h^k - r_h^k - \mathbb{P}_h V_{h+1}^k)(s_h^k, a_h^k)$$
$$+ \underbrace{(\mathbb{P}_h V_{h+1}^k - \mathbb{P}_h V_{h+1}^{\pi^k,k})(s_h^k, a_h^k) - (V_{h+1}^k - V_{h+1}^{\pi^k,k})(s_{h+1}^k)}_{= D_{h,2}^k}, \tag{F.3}$$

where we denote the third term on the right-hand side by $D_{h,2}^k$. Combining (F.2) and (F.3) we obtain

$$\big(V_h^k(s_h^k) - V_h^{\pi^k}(s_h^k)\big) - \big(V_{h+1}^k(s_{h+1}^k) - V_{h+1}^{\pi^k,k}(s_{h+1}^k)\big)$$
$$= D_{h,1}^k + D_{h,2}^k + (Q_h^k - r_h^k - \mathbb{P}_h V_{h+1}^k)(s_h^k, a_h^k). \tag{F.4}$$

Note that $V_{H+1}^k(\cdot) = 0$ for any $k \in [K]$. Using the identity (F.4) for $h \in [H]$ we have

$$V_1^k(s_1^k) - V_1^{\pi^k}(s_1^k)$$
$$= \sum_{h=1}^H \big(V_h^k(s_h^k) - V_h^{\pi^k}(s_h^k)\big) - \big(V_{h+1}^k(s_{h+1}^k) - V_{h+1}^{\pi^k,k}(s_{h+1}^k)\big)$$
$$= \sum_{h=1}^H \big(D_{h,1}^k + D_{h,2}^k + (Q_h^k - r_h^k - \mathbb{P}_h V_{h+1}^k)(s_h^k, a_h^k)\big)$$
$$= \sum_{h=1}^H \big(D_{h,1}^k + D_{h,2}^k\big) + \sum_{h=1}^H \Big(Q_h^k(s_h^k, a_h^k) - \big(r_h^k(s_h^k, a_h^k) + \mathbb{P}_h V_{h+1}^k(s_h^k, a_h^k)\big)\Big).$$

In the sequel, we show that the sequence in (F.1) is a bounded martingale difference sequence with respect to the filtration $\{\widetilde{\mathcal{F}}_t\}_{t \geq 1}$. For any $(k, h) \in [K] \times [H]$, by the definitions of $D_{h,1}^k$ and $D_{h,2}^k$, we have

$$|D_{h,1}^k + D_{h,2}^k| \leq |D_{h,1}^k| + |D_{h,2}^k| \leq 4H.$$

When $h = 1$, we have

$$\mathbb{E}[D_{1,1}^k \,|\, \widetilde{\mathcal{F}}_{i(k,1)-1}] = \mathbb{E}\big[\big(\mathbb{J}_1^k(Q_1^k - Q_1^{\pi^k,k})\big)(s_1^k) - (Q_1^k - Q_1^{\pi^k,k})(s_1^k, a_1^k) \,\big|\, \widetilde{\mathcal{F}}_{i(k-1,H)}\big]$$
$$= \big(\mathbb{J}_1^k(Q_1^k - Q_1^{\pi^k,k})\big)(s_1^k) - \big(\mathbb{J}_1^k(Q_1^k - Q_1^{\pi^k,k})\big)(s_1^k) = 0.$$

Here the second equality is because the only randomness conditional on $\widetilde{\mathcal{F}}_{i(k-1,H)}$ is $a_1^k \sim \pi_1^k(\cdot \,|\, s_1^k)$, which is because, by our definitions of $Q_1^{\pi^k,k}$, $\widetilde{\mathcal{F}}_{i(k-1,H)}$, $Q_1^k$, and $\pi^k$, we have

$$Q_1^k(\cdot), Q_1^{\pi^k,k}(\cdot) \in \widetilde{\mathcal{F}}_{i(k-1,H)}.$$

Similarly, when $h \geq 2$, we have

$$\mathbb{E}[D_{h-1,2}^k \,|\, \widetilde{\mathcal{F}}_{i(k,h)-1}] \tag{F.5}$$
$$= \mathbb{E}[(\mathbb{P}_{h-1}V_h^k - \mathbb{P}_{h-1}V_h^{\pi^k,k})(s_{h-1}^k, a_{h-1}^k) - (V_h^k - V_h^{\pi^k,k})(s_h^k) \,|\, \widetilde{\mathcal{F}}_{i(k,h-1)}]$$
$$= (\mathbb{P}_{h-1}V_h^k - \mathbb{P}_{h-1}V_h^{\pi^k,k})(s_{h-1}^k, a_{h-1}^k) - (\mathbb{P}_{h-1}V_h^k - \mathbb{P}_{h-1}V_h^{\pi^k,k})(s_{h-1}^k, a_{h-1}^k) = 0,$$

which is because the only randomness conditional on $\widetilde{\mathcal{F}}_{i(k,h-1)}$ is $s_h^k \sim P_h(\cdot \,|\, s_{h-1}^k, a_{h-1}^k)$ and we have

$$\mathbb{E}[D_{h,1}^k \,|\, \widetilde{\mathcal{F}}_{i(k,h)-1}] = \mathbb{E}\big[\mathbb{E}[D_{h,1}^k \,|\, \widetilde{\mathcal{F}}_{i(k,h)-1}, s_h^k] \,\big|\, \widetilde{\mathcal{F}}_{i(k,h)-1}\big] = 0. \tag{F.6}$$

Combining (F.5) and (F.6), we obtain

$$\mathbb{E}[D_{h-1,2}^k + D_{h,1}^k \,|\, \widetilde{\mathcal{F}}_{i(k,h)-1}] = 0.$$

Therefore, we conclude the proof of Lemma F.1. $\qquad\square$

**Definition F.2** (Filtration). We define the time index map $i(\cdot, \cdot)$ by

$$i(k, h) = H \cdot (j - 1) + h,$$

for any $(k, h) \in [K] \times [H]$, which is a bijection from $[K] \times [H]$ to $[KH]$. Then, for any $(\tau, h) \in [K] \times [H]$, we define $\widetilde{\mathcal{F}}_{t(k,h)}$ as the $\sigma$-algebra generated by

$$(r^1, s_1^1, a_1^1, \cdots, s_H^1, a_H^1, r^2, s_1^2, a_1^2, \cdots, s_H^{\tau-1}, a_H^{\tau-1}, r^\tau, s_1^\tau, a_1^\tau, \cdots, s_h^\tau, a_h^\tau),$$

when $h \leq H - 1$, which are the reward functions and state-action pairs determined before $s_{h+1}^\tau$, and

$$(r^1, s_1^1, a_1^1, \cdots, s_H^1, a_H^1, r^2, s_1^2, a_1^2, \cdots, s_H^\tau, a_H^\tau, r^{\tau+1}),$$

when $h = H$, which are the reward functions and state-action pairs determined before $s_1^{\tau+1}$. Then, the sequence $\{\widetilde{\mathcal{F}}_t\}_{t \geq 1}$ forms a filtration. Note that $r^\tau = \{r_h^\tau\}_{h=1}^H$ are determined before the beginning of the $\tau$-th episode although they are revealed to the agent until the $\tau$-th episode ends.

## F.2 Concerntration

Let $\{(X_\tau, Y_\tau)\}_{\tau \geq 1}$ be a sequence of random elements in $\mathcal{X} \times \mathbb{R}$ for some measurable set $\mathcal{X}$. Let $\mathcal{Z}$ be a set of $[0, C]$-valued measurable functions with domain $\mathcal{X}$ for some $C > 0$. Let $\mathcal{F} = \{\mathcal{F}_\tau\}_{\tau \geq 1}$ be a filtration such that for any $\tau \geq 1$, $(X_1, Y_1, \cdots, X_{\tau-1}, Y_{\tau-1}, X_\tau)$ is $\mathcal{F}_{\tau-1}$-measurable and there exists $z^* \in \mathcal{Z}$ such that $\mathbb{E}[Y_\tau \,|\, \mathcal{F}_{\tau-1}] = z^*(X_\tau)$ holds. The least-squares predictor $\hat{z}$ given

$\{(X_\tau, Y_\tau)\}_{\tau=1}^t$ is defined by

$$\widehat{z}_t = \underset{z \in \mathcal{Z}}{\operatorname{argmin}} \sum_{\tau=1}^t \big(z(X_\tau) - Y_\tau\big)^2. \tag{F.7}$$

We say that $\eta$ is conditionally $\sigma$-sub-Gaussian given $\mathcal{F}_\tau \in \mathcal{F}$ for any $\tau \geq 1$ if for any $\lambda \in \mathbb{R}$,

$$\log \mathbb{E}[\exp(\lambda \eta) \,|\, \mathcal{F}_\tau] \leq \lambda^2 \sigma^2 / 2.$$

For any $\varepsilon > 0$, we denote by $\mathcal{N}_\varepsilon(\mathcal{Z}, \|\cdot\|_\infty)$ the $\varepsilon$-covering number of $\mathcal{Z}$ with respect to the supremum norm distance $\|z_1 - z_2\|_\infty = \sup_{x \in \mathbb{R}} |z_1(x) - z_2(x)|$. For any $\beta > 0$, we define

$$\mathcal{Z}_t(\beta) = \Big\{ z \in \mathcal{Z} : \sum_{\tau=1}^t \big(z(X_\tau) - \widehat{z}_t(X_\tau)\big)^2 \leq \beta \Big\}. \tag{F.8}$$

**Lemma F.3** (Proposition 6 of Russo & Van Roy (2014))**.** Assume that for any $t \in \mathbb{N}$, $Y_t - z^*(X_t)$ is conditionally $\sigma$-sub-Gaussian given $\mathcal{F}_{t-1}$. Then, for any $\varepsilon > 0$ and $\delta \in [0, 1]$, with probability at least $(1 - \delta)$, it holds that $z^* \in \mathcal{Z}_t(\beta_t(\delta, \varepsilon))$ for any $t \in \mathbb{N}$, where

$$\beta_t(\delta, \varepsilon) = 8\sigma^2 \cdot \log\big(\mathcal{N}_\varepsilon(\mathcal{Z}, \|\cdot\|_\infty)/\delta\big) + 4t\varepsilon\big(C + \sqrt{\sigma^2 \log(4t(t+1)/\delta)}\big).$$

**Lemma F.4.** For any $\delta \in [0, 1]$, if we let

$$\beta \geq 2H^2 \cdot \log\big(\mathcal{N}_{1/(KH)}(\mathcal{P}, \|\cdot\|_{\infty,1}) \cdot H/\delta\big) + 4\big(H + \sqrt{H^2/4 \cdot \log(4K^2 H/\delta)}\big)$$

in Algorithm 1, then, with probability at least $(1 - \delta)$, for any $(k, h) \in [K] \times [H]$, we have $\mathcal{P}_h \in \mathcal{P}_h^k$.

*Proof.* Recall that, for any $p \in \mathcal{P}$, we define $z_P : \mathcal{S} \times \mathcal{A} \times [0, H]^{\mathcal{S}} \to [0, H]$ by

$$z_P\big(s, a, V(\cdot)\big) = \int_{\mathcal{S}} V(s') \cdot P(s' \,|\, s, a) \,\mathrm{d}s', \quad \forall \big(s, a, V(\cdot)\big) \in \mathcal{S} \times \mathcal{A} \times [0, H]^{\mathcal{S}}.$$

Let $\mathcal{Z} = \mathcal{Z}_{\mathcal{P}} = \{z_P : P \in \mathcal{P}\}$. For any $(k, h) \in [K] \times [H]$, we set $Y_k = V_{h+1}^k(s_{h+1}^k)$, $X_k = (s_h^k, a_h^k, V_{h+1}^k(\cdot))$, and $z^* = z_{P_h}$. Then, $Y_\tau - z^*(X_\tau)$ is conditionally $H/2$-sub-Gaussian given $\widetilde{\mathcal{F}}_{i(k,h)}$ defined in Definition F.2. Then, by the definitions of $\mathcal{P}_h^k$ in (3.4) and $\mathcal{Z}_k(\beta)$ in (F.8), we have $\mathcal{Z}_k(\beta) = \{z_P : P \in \mathcal{P}_h^k\}$. By setting

$$\beta \geq 2H^2 \cdot \log\big(\mathcal{N}_{1/K}(\mathcal{Z}) \cdot H/\delta\big) + 4\big(H + \sqrt{H^2/4 \cdot \log(4K^2 H/\delta)}\big)$$

in Algorithm 1, it holds that

$$\beta \geq 2H^2 \cdot \log\big(\mathcal{N}_{1/K}(\mathcal{Z}) \cdot H/\delta\big) + 4(k-1)/K \cdot \big(H + \sqrt{H^2/4 \cdot \log(4k(k-1)H/\delta)}\big)$$
$$= \beta_{k-1}(\delta/H, 1/K)$$

for any $k \in [K]$, where $\beta_{k-1}(\delta/H, 1/K)$ is defined in Lemma F.3. Applying Lemma F.3 with $C = H$, with probability at least $(1 - \delta/H)$, for any $k \in [K]$, we have

$$z^* \in \mathcal{Z}_k\big(\beta_{k-1}(\delta/H, 1/K)\big) \subset \mathcal{Z}_k(\beta),$$

which implies $P_h \in \mathcal{P}_h^k$. Using the union bound over all $h \in [H]$, with probability at least $(1 - \delta)$, for any $(k, H) \in [K] \times [H]$, we have $P_h \in \mathcal{P}_h^k$.

In the sequel, we prove that $\mathcal{N}_\varepsilon(\mathcal{Z}, \|\cdot\|_\infty) \le \mathcal{N}_{\varepsilon/H}(\mathcal{P}, \|\cdot\|_{\infty,1})$ for any $\varepsilon > 0$. Indeed, this is obtained by the observation that, for any $z_P, z_{P'} \in \mathcal{Z}$ with $P, P' \in \mathcal{P}$, we have

$$\|z_P - z_{P'}\|_\infty = \sup_{(s,a,V(\cdot)) \in \mathcal{S} \times \mathcal{A} \times [0,H]^{\mathcal{S}}} \left| \int_{\mathcal{S}} V(s') \cdot P(s' \,|\, s,a) \, \mathrm{d}s' - \int_{\mathcal{S}} V(s') \cdot P'(s' \,|\, s,a) \, \mathrm{d}s' \right|$$

$$\le \sup_{(s,a) \in \mathcal{S} \times \mathcal{A}} H \cdot \int_{\mathcal{S}} |P(s' \,|\, s,a) - P'(s' \,|\, s,a)| \, \mathrm{d}s' = H \cdot \|P - P'\|_{\infty,1}.$$

Thus, we conclude the proof of Lemma F.4. $\qquad\square$

### F.3 ELUDER DIMENSION

Recall that $\mathcal{Z}$ is a set of $[0,C]$-valued functions with domain $\mathcal{X}$ for some $C > 0$. Meanwhile, $\mathcal{Z}_k(\beta)$ is defined in (F.8).

**Lemma F.5** (Lemma 5 of Russo & Van Roy (2014)). *For any $\beta > 0$, we have*

$$\sum_{k=1}^{K} \sup_{z,z' \in \mathcal{Z}_k(\beta)} |z(x_k) - z'(x_k)| \le 1 + C \cdot d + 4 \cdot \sqrt{d\beta K},$$

*where $d = K \wedge \dim_{\mathrm{E}}(\mathcal{Z}, 1/K)$.*

*Proof.* When $\dim_{\mathrm{E}}(\mathcal{Z}, 1/K) \le K$, by Lemma 5 of Russo & Van Roy (2014) we have

$$\sum_{k=1}^{K} \sup_{z,z' \in \mathcal{Z}_k(\beta)} |z(x_k) - z'(x_k)| \le 1 + C \cdot \dim_{\mathrm{E}}(\mathcal{Z}, 1/K) + 4 \cdot \sqrt{\dim_{\mathrm{E}}(\mathcal{Z}, 1/K)\beta K}$$

$$= 1 + C \cdot d + 4 \cdot \sqrt{d\beta K}.$$

When $\dim_{\mathrm{E}}(\mathcal{Z}, 1/K) > K$, since $\mathcal{Z}$ is a set of $[0,C]$-valued functions and $\mathcal{Z}_k(\beta) \subset \mathcal{Z}$ for any $k$ and $\beta$, we have

$$\sum_{k=1}^{K} \sup_{z,z' \in \mathcal{Z}_k(\beta)} |z(x_k) - z'(x_k)| \le KC \le 1 + C \cdot d + 4 \cdot \sqrt{d\beta K}.$$

Thus, we conclude the proof of Lemma F.5. $\qquad\square$

### F.4 OTHER USEFUL INEQUALITIES

**Lemma F.6.** *For any $\gamma \in (0, 1/2)$, $C_2 > 0$, and any $d_0 \in \mathbb{N}$ such that*

$$d_0^\gamma \ge 4(1-\gamma)(\gamma C_2)^{-1}, \tag{F.9}$$

*it holds that*

$$\sum_{j > d_0} \exp(-C_2 j^\gamma/2) \le 4d_0^{1-\gamma}(\gamma C_2)^{-1} \cdot \exp(-C_2 d_0^\gamma/2).$$

*Proof.* By basic calculus we have

$$2d_0^{1-\gamma}/(\gamma C_2) \cdot \exp(-C_2 d_0^\gamma/2) = \left( -2t^{1-\gamma}/(\gamma C_2) \cdot \exp(-C_2 t^\gamma/2) \right)\big|_{t=d_0}^{\infty}$$

$$= \int_{d_0}^{\infty} \left( 1 - 2(1-\gamma)t^{-\gamma}/(\gamma C_2) \right) \exp(-C_2 t^\gamma/2) \, \mathrm{d}t$$

$$\ge \left( 1 - 2(1-\gamma)d_0^{-\gamma}/(\gamma C_2) \right) \cdot \int_{d_0}^{\infty} \exp(-C_2 t^\gamma/2) \, \mathrm{d}t,$$

where the inequality is because $1 - \gamma \geq 0$ and $t^{-\gamma} \leq d_0^{-\gamma}$ for $t \geq d_0$. Thus, we obtain

$$\int_{d_0}^{\infty} \exp(-C_2 t^{\gamma}/2)\, \mathrm{d}t \leq \frac{2d_0^{1-\gamma}/(\gamma C_2)}{1 - 2(1-\gamma)d_0^{-\gamma}/(\gamma C_2)} \exp(-C_2 d_0^{\gamma}/2) \leq 4d_0^{1-\gamma}/(\gamma C_2) \cdot \exp(-C_2 d_0^{\gamma}/2),$$

where the last inequality is because $1 - 2(1-\gamma)d_0^{-\gamma}/(\gamma C_2) \geq 1/2$ by (F.9). Then, by the fact that

$$\sum_{j > d_0} \exp(-C_2 j^{\gamma}) \leq \int_{t=d_0}^{\infty} \exp(-C_2 t^{\gamma})\, \mathrm{d}t,$$

we conclude the proof of Lemma F.6. $\qquad\square$

Recall that $\Gamma(d_0)$ is defined in (C.4).

**Lemma F.7.** Let $C_1$ and $C_2$ be the absolute constants in Assumption 4.2. There exists an absolute constant $\widetilde{C}$ such that for any $\gamma \in (0, 1/2)$, $t \geq 1$, and $R \geq 2$, if we set

$$d_0 = \lceil \widetilde{C} \cdot \log(1/\gamma)/\gamma \cdot \log^{1/\gamma}(tRH) \rceil,$$

then it holds that $d_0^{\gamma} \geq 4(1 - \gamma)(\gamma C_2)^{-1}$ and

$$\Gamma(d_0) = C_1^{1/2} d_0^{1-\gamma} RH(\gamma C_2)^{-1} \cdot \exp(-C_2 d_0^{\gamma}/2) \leq 1/t. \tag{F.10}$$

*Proof.* For any $y > 0$, we consider the function

$$f(x) = e^x/x^y, \; x > 0.$$

Taking derivatives, we have

$$f'(x) = \frac{e^x x^{y-1}(x-y)}{x^{2y}}.$$

Note that $f'(x) \geq 0$ if and only if $x \leq y$, which implies $f(x) \geq f(y)$ for any $x > 0$. Reorganizing the inequality, we obtain

$$e^x \geq (ex/y)^y$$

for any $x > 0$ and $y > 0$. Applying the above inequality via choosing

$$x = C_2 d_0^{\gamma}/4, \quad y = (1-\gamma)/\gamma,$$

we obtain

$$\frac{d_0^{1-\gamma} C_1^{1/2} HR(\gamma C_2)^{-1}}{\exp(C_2 d_0^{\gamma}/2)}$$

$$= \frac{d_0^{1-\gamma}}{\exp(C_2 d_0^{\gamma}/4)} \frac{C_1^{1/2} HR(\gamma C_2)^{-1}}{\exp(C_2 d_0^{\gamma}/4)}$$

$$\leq \frac{d_0^{1-\gamma}}{(eC_2 d_0^{\gamma}/4 \cdot \frac{\gamma}{1-\gamma})^{\frac{1-\gamma}{\gamma}}} \frac{C_1^{1/2} HR(\gamma C_2)^{-1}}{\exp(C_2 d_0^{\gamma}/4)} = \frac{1}{(eC_2/4 \cdot \frac{\gamma}{1-\gamma})^{\frac{1-\gamma}{\gamma}}} \frac{C_1^{1/2} HR(\gamma C_2)^{-1}}{\exp(C_2 d_0^{\gamma}/4)}.$$

Thus, to obtain (F.10), it suffices to make the following inequality hold,

$$\exp(C_2 d_0^{\gamma}/4) \geq \frac{tC_1^{1/2} RH}{\gamma C_2 (eC_2/4 \cdot \frac{\gamma}{1-\gamma})^{\frac{1-\gamma}{\gamma}}},$$

which is equivalent to

$$d_0 \geq \Big( \frac{1}{C_2} \log \frac{t C_1^{1/2} R H}{\gamma C_2 (e C_2/4 \cdot \frac{\gamma}{1-\gamma})^{\frac{1-\gamma}{\gamma}}} \Big)^{1/\gamma}.$$

Since $\gamma \in (0, 1/2)$ and $tRH \geq 2$, there exists an absolute constant $\widetilde{C}$ such that

$$4(1-\gamma)(\gamma C_2)^{-1} \leq \widetilde{C} \cdot \log(1/\gamma)/\gamma \cdot \log(tRH),$$

$$4/C_2 \cdot \log \frac{t C_1^{1/2} H R}{\gamma C_2 (e C_2/4 \cdot \frac{\gamma}{1-\gamma})^{\frac{1-\gamma}{\gamma}}} \leq \widetilde{C} \cdot \log(1/\gamma)/\gamma \cdot \log(tRH).$$

Therefore, by choosing $d_0 \geq (\widetilde{C} \log(1/\gamma)/\gamma \cdot \log(tRH))^{1/\gamma}$, we conclude the proof of Lemma F.7. $\qquad\square$

# G    EXAMPLES OF KERNELS WITH EXPONENTIALLY DECAYING EIGENVALUES

In this section, we provide examples of kernels that satisfies Assumption 4.2. We let $\mathcal{Y} = \mathbb{S}^{d_\mathcal{Y}-1}$, which represents the unit sphere in $\mathbb{R}^{d_\mathcal{Y}}$. For any kernel $\mathcal{K}$, we define the integral operator $\mathcal{T}_\mathcal{K} : L^2(\mathcal{Y}) \to L^2(\mathcal{Y})$ by

$$(\mathcal{T}_\mathcal{K} f)(x) = \int_\mathcal{Y} \mathcal{K}_{\mathrm{se}}(x, y) f(y) \, \mathrm{d}\mu(y), \text{ for any } f \in L^2(\mathcal{Y}) \text{ and } x \in \mathcal{Y} \qquad (\mathrm{G}.1)$$

where $\mu$ is the uniform measure on $\mathcal{Y}$.

## G.1    SQUARED EXPONENTIAL KERNEL

The squared exponential kernel is defined as

$$\mathcal{K}_{\mathrm{se}}(x, y) = \exp\{-1/\iota^2 \cdot \|x - y\|_2^2\}, \text{ for any } x, y \in \mathcal{Y}, \qquad (\mathrm{G}.2)$$

where the constant $\iota$ satisfies $\iota^2 \geq 2/d_\mathcal{Y}$. For any $u \in [-1, 1]$, we define $\widetilde{k}(u) = \exp\{-2(1-u)/\iota^2\}$ and

$$\widetilde{P}_j(u) = \frac{(-1/2)^j \cdot \Gamma((d_\mathcal{Y}-1)/2)}{\Gamma((2j+d_\mathcal{Y}-1)/2)} \cdot (1-u^2)^{(3-d_\mathcal{Y})/2} \cdot \Big( \frac{\mathrm{d}}{\mathrm{d}u} \Big)^j [(1-u^2)^{j+(d_\mathcal{Y}-3)/2}], \qquad (\mathrm{G}.3)$$

where, with a slight abuse of notations, we use $\Gamma(\cdot)$ to denote the Gamma function in this section.

**Lemma G.1** (Theorem 2 of Minh et al. (2006))**.** For the kernel $\mathcal{K}_{\mathrm{se}}$ defined in (G.2), the eigenvalues $\{\rho_j\}_{j \geq 1}$ (without duplicates) of the corresponding integral operator $\mathcal{T}_{\mathcal{K}_{\mathrm{se}}}$ take the form

$$\rho_j = \frac{|\mathbb{S}^{d_\mathcal{Y}-2}|}{|\mathbb{S}^{d_\mathcal{Y}-1}|} \cdot \int_{-1}^1 \widetilde{k}(u) \cdot \widetilde{P}_j(u; d_\mathcal{Y}) \cdot (1-u^2)^{(d_\mathcal{Y}-3)/2} \, \mathrm{d}u,$$

and each $\rho_j$ has multiplicity

$$N(j) = \frac{(2j+d_\mathcal{Y}-2) \cdot (d_\mathcal{Y}+j-3)!}{j!(d_\mathcal{Y}-2)!}. \qquad (\mathrm{G}.4)$$

Moreover, when $\iota^2 \geq 2/d_\mathcal{Y}$, we have that $\{\rho_j\}_{j \geq 1}$ is in a decreasing order and satisfies

$$\rho_j > A_1 \cdot \Big( \frac{e}{\iota^2} \Big)^j \cdot (2j+d_\mathcal{Y}-2)^{-(2j+d_\mathcal{Y}-1)/2},$$

$$\rho_j < A_2 \cdot \Big( \frac{e}{\iota^2} \Big)^j \cdot (2j+d_\mathcal{Y}-2)^{-(2j+d_\mathcal{Y}-1)/2}, \qquad (\mathrm{G}.5)$$

for any $j \geq 1$, where the constants $A_1, A_2$ only depend on $d_{\mathcal{Y}}$ and $\iota$.

By Lemma G.1, the eigenvalues $\{\lambda_j\}_{j \geq 1}$ (with duplicates) of the kernel $\mathcal{K}_{\text{se}}$ satisfy, for any $j \geq 1$,

$$\lambda_j = \rho_t, \text{ for } t \text{ such that } \sum_{i=1}^{t-1} N(i) < j \leq \sum_{i=1}^{t} N(i). \tag{G.6}$$

By the definition of $N(j)$ in (G.4) and Stirling's formula, we have

$$N(j) \asymp \frac{(2j + d_{\mathcal{Y}} - 2) \cdot (d_{\mathcal{Y}} + j - 3)^{1/2} \cdot [(d_{\mathcal{Y}} + j - 3)/e]^{(d_{\mathcal{Y}}+j-3)}}{j^{1/2} \cdot (j/e)^j} \asymp j^{d_{\mathcal{Y}}-2}. \tag{G.7}$$

Here the asymptotic notation $\asymp$ omits constant factors that are independent of $j$. Combining (G.6) and (G.7), when $j$ is sufficiently large, we have

$$\lambda_j = \rho_t, \text{ for } t \text{ such that } (t-1)^{d_{\mathcal{Y}}-1} < j \leq t^{d_{\mathcal{Y}}-1}.$$

Then, by (G.5) we obtain

$$\lambda_j = O\left(\left(\frac{e}{\iota^2}\right)^{j^{1/d_{\mathcal{Y}}}} \cdot (2j^{1/d_{\mathcal{Y}}} + d_{\mathcal{Y}} - 2)^{-(2j^{1/d_{\mathcal{Y}}}+d_{\mathcal{Y}}-1)/2}\right) = O(e^{-c \cdot j^{1/d_{\mathcal{Y}}}}) \tag{G.8}$$

as $j \to \infty$ for an absolute constant $c > 0$. Thus, we know $\mathcal{K}_{\text{se}}$ satisfies the second condition of Assumption 4.2.

## G.2 NTK OF SINE ACTIVATION

We consider the neural tangent kernel of a two-layer neural network of the form (E.1) where the activation function is the sine function. In detail, the neural network is parametrized as

$$\text{NN}(x; w, l) = \sqrt{\frac{2}{m/(d_{\mathcal{Y}}+1)}} \cdot \sum_{j=1}^{m/(d_{\mathcal{Y}}+1)} b_j \cdot \sin(x^\top w_j + l_j)$$

for any $x \in \mathcal{Y}$. Here we modify the initial form in (E.1) by adding an intercept term, which is equivalent to adding one more dimension with constant value 1 to the input space. The initialization of the network weights follows the same symmetric random initialization scheme

$$b_j \overset{\text{i.i.d.}}{\sim} \text{Unif}(\{-1, 1\}), \quad b_{j+m/(2d_{\mathcal{Y}}+2)} = b_j,$$
$$w_j \overset{\text{i.i.d.}}{\sim} \mathcal{N}(0, I_{d_{\mathcal{Y}}}), \quad w_{j+m/(2d_{\mathcal{Y}}+2)} = w_j,$$
$$l_j \overset{\text{i.i.d.}}{\sim} \text{Unif}([0, 2\pi]), \quad l_{j+m/(2d_{\mathcal{Y}}+2)} = l_j,$$

for $j \in [m/(2d_{\mathcal{Y}} + 2)]$. Here without loss of generality we assume $m/(2d_{\mathcal{Y}} + 2) \in \mathbb{N}$. Then, the population NTK of such a parametrization takes the form

$$\begin{aligned}
\mathcal{K}_{\text{ntk}}(x, y) &= 2 \cdot \mathbb{E}_{W \sim I_{d_{\mathcal{Y}}}, L \sim \text{Unif}([0,2\pi])}[x^\top y \cdot \cos(x^\top W + L) \cdot \cos(y^\top W + L)] \\
&= x^\top y \cdot \exp\{-\|x - y\|_2^2/2\} \\
&= x^\top y \cdot \exp\{x^\top y - 1\},
\end{aligned}$$

for any $x, y \in \mathcal{Y}$, which is the limit of the empirical NTK defined in (4.2) as $m$ goes to infinity. Here the second equality is derived in Rahimi & Recht (2007) and the third equality is by the fact that $\|x\|_2 = \|y\|_2 = 1$ for any $x, y \in \mathcal{Y} = \mathbb{S}^{d_{\mathcal{Y}}-1}$.

For any $j \geq 1$, let $\mathcal{Y}_j$ be the set of all homogeneous harmonics of degree j on $\mathbb{S}^{d_{\mathcal{Y}}-1}$, which is a finite-dimensional subspace of $L_\mu^2(\mathbb{S}^{d_{\mathcal{Y}}-1})$, the space of square-integrable functions on $\mathbb{S}^{d_{\mathcal{Y}}-1}$ with respect to $\mu$. It can be shown that the dimensionality of $\mathcal{Y}_j$ is given by $N(j)$.

**Lemma G.2** (Funk-Hecke formula (Müller (2012), page 30))**.** Let $\widetilde{k}_2 : [-1, 1] \to \mathbb{R}$ be a continuous function, which gives rise to an inner product kernel $\widetilde{\mathcal{K}}$ on $\mathbb{S}^{d_\mathcal{Y}-1} \times \mathbb{S}^{d_\mathcal{Y}-1}$ with the definition

$$\widetilde{\mathcal{K}}(x, y) = \widetilde{k}_2(x^\top y), \text{ for any } x, y \in \mathbb{S}^{d_\mathcal{Y}-1}.$$

Then, for any $j \geq 1$, $f \in \mathcal{Y}_j$, $x \in \mathbb{S}^{d_\mathcal{Y}-1}$, we have

$$\int_{\mathbb{S}^{d_\mathcal{Y}-1}} \widetilde{\mathcal{K}}(x, y) f(y) \, \mathrm{d}\mu(y) = \left( \frac{|\mathbb{S}^{d_\mathcal{Y}-2}|}{|\mathbb{S}^{d_\mathcal{Y}-1}|} \cdot \int_{-1}^{1} \widetilde{k}_2(u) \cdot \widetilde{P}_j(u) \cdot (1 - u^2)^{(d_\mathcal{Y}-3)/2} \, \mathrm{d}u \right) \cdot f(x),$$

where $\widetilde{P}_j(u)$ is defined in (G.3).

We let $\widetilde{k}_2(u) = u \cdot \exp\{u - 1\}$. Recall the definition of $\widetilde{k}$ in Section G.1, we have $\widetilde{k}_2(u) = u \cdot \widetilde{k}(u)$ for $\iota = \sqrt{2}$, which satisfies the requirement in Lemma G.1. Lemma G.2 shows that the eigenvalues $\{\widetilde{\rho}_j\}_{j \geq 1}$ (with duplicates) of $\mathcal{T}_{\widetilde{\mathcal{K}}}$ takes the form

$$\widetilde{\rho}_j = C_\rho \cdot \int_{-1}^{1} \widetilde{k}_2(u) \cdot \widetilde{P}_j(u) \cdot (1 - u^2)^{(d_\mathcal{Y}-3)/2} \, \mathrm{d}u$$

$$= C_\rho \cdot \int_{-1}^{1} u \cdot \widetilde{k}(u) \cdot \widetilde{P}_j(u) \cdot (1 - u^2)^{(d_\mathcal{Y}-3)/2} \, \mathrm{d}u,$$

where $C_\rho = |\mathbb{S}^{d_\mathcal{Y}-2}|/|\mathbb{S}^{d_\mathcal{Y}-1}|$. Using the relation

$$u \cdot \widetilde{P}_j(u) = \frac{j}{2j + d_\mathcal{Y} - 2} \cdot \widetilde{P}_{j-1}(u) + \frac{j + d_\mathcal{Y} - 2}{2j + d_\mathcal{Y} - 2} \cdot \widetilde{P}_{j+1}(u),$$

which is from the definition of $\widetilde{P}_j(u)$, we have

$$\widetilde{\rho}_j = C_\rho \cdot \left( \frac{j}{2j + d_\mathcal{Y} - 2} \cdot \rho_{j-1} + \frac{j + d_\mathcal{Y} - 2}{2j + d_\mathcal{Y} - 2} \cdot \rho_{j+1} \right),$$

where $\{\rho_j\}_{j \geq 1}$ are the eigenvalues of the operator $\mathcal{T}_{\mathcal{K}_{\mathrm{se}}}$ studied in Section G.1 with $\iota = \sqrt{2}$. Thus, following the same argument of (G.6)-(G.8), we know such an NTK satisfies the second condition of Assumption 4.2.

