# OpenReview forum: "Optimistic Policy Optimization with General Function Approximations"
_ICLR.cc/2021/Conference — Reject_

### Official Review · AnonReviewer2 · 2020-10-28
**Solid theoretical results on policy optimization with function approximation**

**Rating:** 7
**Confidence:** 4

**Review:**

The paper proposes an optimistic policy optimization algorithm. It is theoretically shown that the algorithm has sublinear regret for multiple model classes such as kernel function and NTK. The policy update rule also allows the algorithm to have sublinear regret for  adversarial reward function. The technique for analyzing nonparametric model classes can potentially be extended to other reinforcement learning algorithms based on optimism.

While the algorithm itself is neat and clean, the paragraph describing its implementation is not very rigorous. In particular, how is Line 13 computed? Note that for general value function approximation, the state space is usually large (or even infinite).  If I understand correctly, there is no assumption about the structure of Q-function in this paper. How is the Q-function even stored, or parameterized?

A similar question is that, the last few sentences before Sec 4 says "it suffices to solve a least-square regression problem". Does it mean that $Q_h^k(\cdot\mid\cdot)$ is computed by doing a regression problem where the target is r+clamp...? If this is the case, can the Q-function be solved exactly? Can the algorithm work for the case where the regression error is small but not zero?

It seems to me that the analysis for general function class is different to those for kernel function and NTK, so I'm wondering that are there any upper/lower bounds for the Eluder dimension of the two non-parametric classes described in this paper? Besides that, the regret upper bound given by Theorem 4.3 and 4.7 depends on parameter $\gamma$. In what scenario the parameter $\gamma$ is lower bounded by some constant?

Minor issue:
- In Eq. (3.1), Page 3: should the policy be $\pi_h^k(a\mid s)=\frac{\exp(E_h^k(s,a))}{\sum \exp(E_h^k(s,a))}$?

Overall, the results are solid and interesting. Therefore I would recommend a acceptance.

---

> ### Author Response · Authors · 2020-11-19
> **Reply to Reviewer 2**
>
> We appreciate the valuable review and suggestions. We have revised our work accordingly.
>
> In what follows, we address your concerns in detail.
>
> 1.(implementation of the algorithm) Yes, we will need another parametrization to store the updated Q-functions, which will be trained through least square regressions to fit the target. We note that in such a regression problem, we have as much training data as we need, and thus the error can usually be reduced to any sufficient accuracy, for example, using a non-parametric model. The computational error, if considered, can be trivially separated as a part of the regret and will be dominated by the terms in our main theorems given enough computational power.
>
> 2.(typos) Yes, equation (3.1) should be written in that form. We have corrected it in the revised pdf.

---

### Official Review · AnonReviewer3 · 2020-10-28
**Recommendation to Accept**

**Rating:** 6
**Confidence:** 1

**Review:**

This paper propose a novel policy optimization algorithm that allows general function approximation (i.e., kernel function approximation, neural function approximation) that go beyond linear approximation, the algorithm achieves exploration by absorbing optimism into policy evaluation. An \sqrt{T} regret bound of the proposed algorithm is obtained.

Overall, this paper is very well-written, the problem is well-motivated, and the claims and proofs looks solid.

---

> ### Author Response · Authors · 2020-11-19
> **Reply to Reviewer 3**
>
> We appreciate your review of our work. We have addressed the issues raised by the other reviewers and revised our work accordingly.

---

### Official Review · AnonReviewer4 · 2020-10-28

**Rating:** 5
**Confidence:** 4

**Review:**

**Summary:**

This paper proves a regret bound for an optimistic variant of a policy optimization algorithm in an advsersarial reward setting. The paper extends prior work in this setting by considering function classes with bounded Eluder dimension instead of linear functions. This yields guarantees for a kernel-based variant of the algorithm (which also apply to neural kernels like the NTK).

--------------------------------------------------------------------

**Strengths:**

1. More general than prior work. The main strength of this paper is to extend the work of Cai et al. (2019) beyond the Linear MDP setting to one with a kernelized transition matrix. This essentially requires introducing some results based on the Eluder dimension from Russo and Van Roy (2014). The paper helps to fill out this line of research by showing that the algorithm from Cai et al. (2019) does indeed work when the linear MDP is replaced by a kernelized one.
2. The proofs seem to be correct.

--------------------------------------------------------------------

**Weaknesses:**

1. Setting seems unrealistic. The paper assumes that the *entire reward function* is revealed after each episode rather than only seeing the rewards at the visited states. This is integral to the algorithm since this reward function is then queried at new state, action pairs. I understand that this allows for adversarial reward functions and is the same assumption from Cai et al. (2019), but it seems unrealistic and unmotivated. I cannot think of a problem where the *entire reward function* would be revealed at the end of each episode, and without this assumption the entire algorithm and argument seem to break down.
2. Calling this algorithm policy optimization seems misleading. The algorithm requires using the kernel-based approximator to learn not just a model, but an entire confidence set over models. In the usual RL nomenclature this would seem to be a model-based algorithm. While the policy is not simply attained by planning in the learned model, having to learn these models seems to make the algorithm indeed model-based. This would not be so important except a major motivation for the paper in the abstract and intro is to provide guarantees for model-free policy optimization algorithms like the policy gradient algorithms that are often used in practice. This motivation does not align with the algorithm the paper is actually analyzing.
3. The main assumption is not clearly explained. The main assumption in the paper is that the transition kernel is in some RKHS, but the paper never explains what this assumption means and which sorts of MDPs it may apply to. It is more general than the Linear MDP, but how much more general? Moreover, the introduction makes it sound like the function approximation assumption is merely for the policy class, but in fact the approximator must be able to represent the entire transition kernel, which is potentially much more complicated.
4. Novelty. To me the paper is only an incremental improvement over the results of Cai et al. (2019). Specifically, extending results from a linear setting to a kernel setting provides a technical challenge, but it is not clear how much insight is gained. The main result follows quickly from combining the results on Eluder dimension from Russo and Van Roy (2014) with thos of Cai et al. (2019).
5. Algorithm seems computationally infeasible. The algorithm requires maximizing an integral over the entire state space against over a set of measures defined by a confidence set over models in the RKHS at every step. The paper argues that the algorithm can be implemented with supervised learning oracles, but having to solve several supervised learning problems at each step of training over all historical data seems quite inefficient. In short, this is not an algorithm that a practitioner would attempt to implement.

--------------------------------------------------------------------

**Recommendation:**

I recommend to reject this paper, and gave it a score of 4. While the paper is a logical extension of some prior work, I am not convinced of the usefulness of the setting or algorithm proposed.

If the authors can provide better motivation for the setting, novelty, and practical value of the contribution, I would consider raising my score.

---

> ### Author Response · Authors · 2020-11-19
> **Reply to Reviewer 4**
>
> We appreciate the valuable review and suggestions. We have revised our work accordingly.
>
> In what follows, we address your concerns in detail.
>
> 1.(unrealistic reward function setting) The instant reward function in reinforcement learning is usually known or defined by human in the modeling phase, and thus it is quite common to assume it to be known. Actually, in practice, oftentimes it is up to the learner to design a reward function, which is known as the reward shaping problem. After the reward is designed, a reinforcement learning problem is solved with that reward. As a concrete example, in many Atari games, reward functions are directly given by the rules and are usually trivial indicator functions of the states.
>
> In our setting, since we consider adversarial rewards, which can depend on previous actions, we assume the revealing of the reward function is delayed. Moreover, our algorithm and proof can be easily adapted to the case of stationary reward function with stochastic feedback. For such a case, we can directly estimate the reward function via least-squares regression and add another bonus function in the policy evaluation step to address the uncertainty in reward function estimation. Moreover, it is possible to allow stochastic feedbacks under the adversarial reward setting. We can also construct confidence sets based on estimates of the reward function using techniques from the bandit literature, e.g., [3].
>
> Furthermore, we would emphasize that the core difficulty of reinforcement learning is to make optimal sequential decisions with the transition model unknown. Such a challenge is known as ``deep exploration’’ [4] which means that an exploration algorithm needs to consider the effect of any action on future learning, which is propagated by the unknown transition model. Even when the reward function is known, such a fundamental challenge persists due to the unknown transition. This is the fundamental difference from the bandit setting, where exploration only focuses on the immediate effect of an action.
>
> Thus, even though we assume the reward function is known, such a setting is realistic from both practical and theoretical perspectives.
>
>
> 2.(name of policy optimization) The algorithm of the paper is model-based, and we have made it clear in the abstract and introduction. We would like to emphasize that the model-free versus model-based classification of RL methods is orthogonal to the policy-based versus value-based algorithm. An algorithm is policy-based simply means that the algorithm updates the policy directly by an optimization method (e.g. policy gradient), and the update direction can be calculated by either model-based or model-free methods. Our algorithm is policy-based because the algorithm constructs a sequence of policies by KL-divergence regularized optimization, which is a local greedy update similar to the natural policy gradient method. This algorithm is model-based because the descent direction, namely the critic, is obtained by model-based policy evaluation. See also Point 1 for Reviewer 5.
>
> In the paper, we emphasize that our algorithm is a policy optimization method in order to distinguish it from purely value-based methods, where the learned policy is always the greedy policy with respect to the value function, and these methods update the policy indirectly by updating the value functions instead. In contrast, policy optimization essentially searches policies within the stochastic policy space.
>
>
> 3.(assumption of transition kernel in some RKHS) RKHS is widely used in machine learning and known as a rich class of functions, which covers the linear function as a special case by choosing the kernel to be the inner product kernel. As a result, our RKHS transition kernel covers the class of linear transitions as a special case.
>
> Moreover, note that the inner product kernel (the kernel of the linear case) is of finite-rank. Thus, the transition kernels expressed by linear class is only finite-dimensional functions. In contrast, by considering RKHS functions in general, we cover __infinite-dimensional__ transition functions. Thus, the model considered in our work is __strictly much more general__ than that in [1]. One example is the transition kernel can be represented using the random feature model [5] induced by a given kernel, which is also closely related to the case of NTK in our paper. Such a class of functions is infinite-dimensional and cannot belong to the finite-dimensional linear model.

---

> > ### Author Response · Authors · 2020-11-19
> > **Reply to Reviewer 4 - continued**
> >
> > In addition, we note that in the proof of [1], each individual feature contributes to the regret upper bound, which makes the regret upper bound linearly depends on the number of features. Thus, the theory in [1] cannot be directly applied to the RKHS setting and a novel regret analysis is required. Moreover, in terms of algorithm implementation, the linear model also requires prior knowledge of the feature mapping, which oftentimes requires nontrivial feature engineering. In contrast, RKHS offers additional flexibility in the sense that the algorithm only requires the knowledge of a reproducing kernel.
> >
> > In sum, the superiority of RKHS over finite-dimensional linear models is three-fold: (1). It contains a strictly larger function class that is allowed to be infinite-dimensional, (2). the study of RKHS model leads to a more general analysis framework as given in this work, and (3). It offers flexibility in algorithm design.
> >
> > (parameterization of the policy class) In our paper, the function approximation assumption is imposed on the transition kernel instead of on the policy class directly. The policy is parameterized via an energy-based form. The parameterized transition model is then used to guide the update of the policy parametrization via the KL-divergence regularized step. We have made it clear in the introduction that our algorithm is model-based and the function approximation assumption is imposed on the transition model.
> >
> >
> > 4.(Novelty) We first note that applying the eluder framework [6] to the **policy-based optimization** approach in MDPs itself is a novel practice, as it has never been shown doable before. Previous related works only consider purely value-based approaches, where parametrizations are used to learn the **optimal value function**. The value-based approach and our policy-based approach is fundamentally different. The main difference lies in that value-based methods directly estimate the optimal value function and the policy is always the greedy policy with respect to the estimated value function, while policy-based optimization learns the value-function associated with the current policy and uses the estimated value function to update the policy. As such, the sequence of estimated value function appear in policy optimization can be much worse than the optimal value function, while the previous optimistic value-based methods always maintain optimistic estimates of the optimal value functions. Thus, it is unclear how to directly apply the eluder dimension framework to policy optimization to handle general function approximation. We cannot simply combine the results for value-based methods with eluder dimension [6] to policy optimization and obtain a novel algorithm.
> >
> > In addition, previous works on eluder dimension only analyze the eluder dimension of finite-dimensional linear models or generalized linear models with very limited link functions. Thus, powerful function approximators such as infinite-dimensional RKHS functions and over-parameterized neural networks are **not** included in previous works. Besides, if we neglect the eluder dimension framework and directly extend the result in [1] to infinite-dimensional RKHS, since the regret in [1] depends linearly on the dimension of the feature mapping, such an approach leads to a vacuous bound. Therefore, directly extending the theory in [1] or combine the theory of [1] and [6] cannot lead to the desired result.
> >
> > In this work, we for the first time establish such a policy optimization algorithm that provably explores under the general function approximation setting. Our algorithm adopts a model-based policy evaluation step that is novel. Moreover, to analyze this algorithm, we provide a general theoretical framework that bridges policy optimization and general function approximation under the eluder dimension framework and establish a regret bound. Our theory covers the result in [1] as a special example while also cover powerful function approximators such as RKHS functions and overparameterized neural networks. To this end, we further enrich the theory of the eluder dimension and demonstrate the power of our theory framework by establishing the eluder dimension of RKHS, and overparametrized neural networks.
> >
> > In terms of technical contribution, more importantly, our paper is the first that incorporates RKHS (and the NTK regime of overparametrized neural networks) into the eluder framework, which is not studied before even for bandits. Such a novel eluder theory is not only limited to policy optimization but also can contribute to the theoretical study of other RL algorithms with kernel or neural network function approximation including UCRL and model-based value iteration.

---

> > > ### Author Response · Authors · 2020-11-19
> > > **Reply to Reviewer 4 - continued**
> > >
> > > 5.(computational efficiency) (1) The computational inefficiency is a cost of the statistical efficiency as the algorithm intends to thoroughly utilize the information from each new observation to guide further explorations. When the environment is in the real world instead of a simulator in the computer, such a computational cost is worthy compared with the expansive exploration cost. (2) Doing an individual computation step based on all historical data before each new interaction is standard in almost all online learning (regret minimization) problems. In the linear or kernel setting, the Lagrangian relaxation of the constrained problem actually has analytical solutions ([1], [2]). In a general setting, this step requires a supervised learning oracle, which is inevitable but applies to any complex model. (3) Our algorithm proposes a general framework for optimistic policy optimization, and we prove its statistical efficiency. We hope our result can serve as the first step of constructing RL algorithms that are both computationally and statistically efficient. It leaves for follow-up work to develop more practical algorithms that are computationally efficient and do not lose too much essence of the current algorithm. This is just like before quasi-Newton methods are developed, people first study how Newton’s method can accelerate convergence.
> > >
> > > Reference:
> > >
> > > [1] Cai, Q., Yang, Z., Jin, C. and Wang, Z., 2019. Provably efficient exploration in policy optimization. arXiv preprint arXiv:1912.05830.
> > >
> > > [2] Chowdhury, S.R. and Gopalan, A., 2017. On kernelized multi-armed bandits. arXiv preprint arXiv:1704.00445.
> > >
> > > [3] Russac, Y., Vernade, C. and Cappé, O., 2019. Weighted linear bandits for non-stationary environments. arXiv preprint arXiv:1909.09146
> > >
> > > [4] Osband, I., Van Roy, B., Russo, D.J. and Wen, Z., 2019. Deep Exploration via Randomized Value Functions. Journal of Machine Learning Research, 20(124), pp.1-62.
> > >
> > > [5] Rahimi, A. and Recht, B., 2007. Random features for large-scale kernel machines. Advances in neural information processing systems, 20, pp.1177-1184.
> > >
> > > [6] Russo, D. and Van Roy, B., 2013. Eluder dimension and the sample complexity of optimistic exploration. In Advances in Neural Information Processing Systems (pp. 2256-2264).

---

> > > > ### Comment · AnonReviewer4 · 2020-11-23
> > > > **response**
> > > >
> > > > Thank you for the thorough response. While the response resolves some issues, I think that the main issues that I raised in my review are still present in the updated version. I will upgrade my score to a 5 to reflect the novelty of the Eluder analysis in the RKHS setting, but still think the fundamental problems with the setup limit the impact of the paper. Below I explain my reasoning on each issue.
> > > >
> > > > 1. I agree that in practical settings the reward function is often known. But in those cases the reward function is clearly not adversarial. The issue with the presented setting is that the reward function is somehow both known and adversarial, which does not seem to make sense. The results still apply in the known, non-adversarial setting, which seems practical. The emphasis in the paper on the known and adversarial setting seems misplaced.
> > > > 2. Thank you for adding the clarification to the abstract and intro that the algorithm is model-based. While these sort of semantic distinctions are not always very clear, in my mind the three classes of algorithms are determined by which object is learned directly from the data: a policy, value, or model, yielding policy-based, value-based, and model-based algorithms. As an example, every paper cited in the first paragraph of the introduction uses a *model-free* policy optimization algorithm, which corroborates that the motivation in studying policy optimization is in fact understanding the model-free version. By learning a model of the environment, the algorithm proposed in the paper is essentially a model-based algorithm that performs planning in the model using a policy and Q function rather than just a Q function. It is not clear that an algorithm like this has anything to say about the model-free algorithms that seem to motivate the paper.
> > > > 3. I understand that the RKHS is more general than the so-called Linear or Low-rank MDP. However, it is still unclear how general it is. Some illustrative examples of useful MDPs that fall within this framework would help.
> > > > 4. I agree that the paper cannot be recovered directly from Cai et al. (2019) and provides the extension from linear to RKHS which is nontrivial. The combination of RKHS with an Eluder dimension analysis indeed seems to be novel.  However, the issue that Reviewer 5 and I are raising is that it is not clear that the extension provides much insight beyond the paper of Cai et al. (2019). Especially since the algorithm is truly model based, it is not so surprising that an Eluder dimension analysis as in Ayoub et al. (2020) can be extended to this setting.
> > > > 5. It is a good point that it is useful to prove statistical efficiency first and hope to develop better algorithms later. So I think this issue is resolved.

---

> > > > > ### Author Response · Authors · 2020-11-24
> > > > > **Reply to Reviewer 4**
> > > > >
> > > > > We thank the reviewer for the response. We address the detailed comments separately as follows.
> > > > >
> > > > > 1.(reward function) First we would like to thank the reviewer for agreeing that it is often the case that the reward function is known in practice. We would like to emphasize that our algorithm can be readily applied to the case where the reward is non-adversarial and either known or unknown.
> > > > >
> > > > > There are several reasons for focusing on such a setting with known and adversarial reward functions.
> > > > >
> > > > > First, this setting theoretically demonstrates the potential advantage of policy optimization methods over value-based methods. Specifically, when the reward function is adversarially chosen, it is possible for the malicious adversary to choose a sequence of reward functions such that the optimistic value-iteration or UCRL incur a $\Omega(T)$ regret. In contrast, our algorithm achieves a sublinear $O(\sqrt{T})$ regret. Note that all of these methods achieve $O(\sqrt{T})$ regret under the static regret setting. By focusing on the adversarial reward setting, we can have a fine-grained comparison between these algorithms and understand the advantage of policy-based optimization.
> > > > >
> > > > > Second, even in practice, such an adversarial reward setting is meaningful. This setting covers the case where the reward functions are time-varying, which is very common in practice. Moreover, this setting also covers the case of domain randomization and adversarial training, where we gradually train the RL agent to tackle more and more challenging tasks that share the same transition model. Consider the domain randomization problem where the RL agent is provided by a sequence of tasks where only the reward functions differ, our algorithm directly implies that it finds the best policy in hindsight. Whereas such a result cannot be obtained by value-based methods without significantly modifying the algorithm.
> > > > >
> > > > > Third, the setting is a good fit for some operations research problems,
> > > > > where, for example, you are maintaining stock to serve some demand and the unknown aspects (besides demands) are the prices, which determine the reward. In these settings, one puts in an order to restock, and the order is valid at the momentary market price, which is unknown at the time the order is put in but becomes known at the end of the period. Once the price is known, one can retrospectively calculate the reward for any action. There are similar examples in control, where, e.g., a setpoint which is set externally needs to be tracked. Once the setpoint becomes known, the rewards are fully known.
> > > > >
> > > > >
> > > > > 2.(model-based) We would like to emphasize that the trichotomy of RL methods into value-based, policy-based, and model-based is inaccurate. In addition to policy-based and model-based methods as the one in our paper and [4], there are also model-based and value-based methods (see, e.g., [5] and [6]). Moreover, all of these two types of algorithms have been studied in both theory and practice. Thus, we would like to point out that policy-based and value-based, and model-free and model-based are two orthogonal dimensions. We provide several examples of all combinations in the following table.
> > > > >
> > > > > |                        Policy/Value                       |    Model    |    Examples    |
> > > > > |:---------------------------------------------------------:|:-----------:|:--------------:|
> > > > > |                        Value-based                        |  Model-free |   DQN, SARSA   |
> > > > > |                                                           | Model-based |  [3], [5], [6] |
> > > > > |          Policy-based (without policy evaluation)         |  Model-free |    REINFORCE   |
> > > > > |                                                           | Model-based |      MCTS      |
> > > > > | Policy-based (with policy evaluation, i.e., actor-critic) |  Model-free | SAC, TRPO, PPO |
> > > > > |                                                           | Model-based | [4], our paper |

---

> > > > > > ### Author Response · Authors · 2020-11-24
> > > > > > **Reply to Reviewer 4 - continued**
> > > > > >
> > > > > > In addition, we would like to point out that the reviewer's argument that “every paper cited in the first paragraph of the introduction uses a model-free policy optimization algorithm, which corroborates that the motivation in studying policy optimization is in fact understanding the model-free version” seems not completely correct. The paper [Baxter & Bartlett, 2000] studies policy gradient with a given parametrized model. The papers we cited in the first paragraph only contain the pioneering works of policy gradient, namely [Williams, 1992; Baxter & Bartlett, 2000; Sutton et al., 2000], as well as those that are most close to our KL-regularized policy update, namely [Kakade, 2002; Schulman et al., 2015; 2017]. We do not intend to provide a thorough survey of policy gradient methods, which is obviously beyond the scope of our paper.
> > > > > >
> > > > > > Nevertheless, we would like to mention that model-based policy optimization has its own merits from both **practical** and **theoretical** perspectives. For example, in robotic applications, it is commonly the case that there is a physical model that contains some unknown parameters. When this is the case, directly estimating the model instead of discarding such important prior knowledge might be the key to achieving sample efficiency. Besides, in terms of theory, model-based policy optimization is also more sample efficient.
> > > > > >
> > > > > > Moreover, the reviewer is right in that the semantic distinctions are not always clear. In fact, if an algorithm keeps all the data, e.g., LSVI-UCB,
> > > > > > would we call this a model-based or a model-free algorithm? In fact, the recent paper of Gergely Neu and Ciara Pike-Burke [7] explicitly shows that what people considered in the past model-based or model-free can be transformed into each other in some special cases. One specific view is that an algorithm is model-based if it comes up with a model that is used by a planning algorithm to come up with a policy that is supposed to be a good policy for the model. Our algorithm does not do this, but the model-class (and not a particular model) is used by it to come up with an optimistic value function for the current policy. It is this viewpoint that we followed that made us consider our algorithm closest to model-free (direct) methods. While we find this discussion interesting and we will be more than happy to add a detailed analysis, it seems to us that confusion about terminology is something that is relatively easy to address and as such should not play a major role in evaluating the merits of a work.
> > > > > >
> > > > > > 3.(RKHS) Any MDP with sufficiently smooth transition kernels satisfies the RKHS condition. For example, mechanical systems (e.g., robots) should fall into this category while simple linear models are unable to parametrize such systems. Besides, overparametrized neural networks have shown their strong expressive power (even with regularization on the weights), which implies that the RKHS ball of an NTK is a good example of large transition kernel spaces. In addition, the random feature model is also a common function approximation tool used in practice and when the number of random features converges to infinity, the function space becomes an RKHS.
> > > > > >
> > > > > >
> > > > > > 4.(novelty and importance of the result) The target of the paper is to extend the $O(\sqrt{T})$ regret result in [1] to the case of infinite-dimensional and non-linear models, which is rarely available in the literature of RL, including all other RL approaches. The eluder dimension framework serves as a tool in our analysis, based on which we make nontrivial efforts to prove the aforementioned case. Thus, our result is clearly a big step beyond [1] because the low-rank linear model in [1] is very limited and the construction of the bonus function in [1] only applies to the linear model.

---

> > > > > > > ### Author Response · Authors · 2020-11-24
> > > > > > > **Reply to Reviewer 4 - continued**
> > > > > > >
> > > > > > > Reference
> > > > > > >
> > > > > > > [1] Cai, Q., Yang, Z., Jin, C. and Wang, Z., 2019. Provably efficient exploration in policy optimization. arXiv preprint arXiv:1912.05830.
> > > > > > >
> > > > > > > [2] Russo, D. and Van Roy, B., 2013. Eluder dimension and the sample complexity of optimistic exploration. In Advances in Neural Information Processing Systems (pp. 2256-2264).
> > > > > > >
> > > > > > > [3] Ayoub, A., Jia, Z., Szepesvari, C., Wang, M. and Yang, L.F., 2020. Model-Based Reinforcement Learning with Value-Targeted Regression. arXiv preprint arXiv:2006.01107.
> > > > > > >
> > > > > > > [4] Janner, M., Fu, J., Zhang, M. and Levine, S., 2019. When to trust your model: Model-based policy optimization. In Advances in Neural Information Processing Systems (pp. 12519-12530).
> > > > > > >
> > > > > > > [5] Farahmand, A.M., 2018. Iterative value-aware model learning. In Advances in Neural Information Processing Systems (pp. 9072-9083).
> > > > > > >
> > > > > > > [6] Tamar, A., Wu, Y., Thomas, G., Levine, S. and Abbeel, P., 2016. Value iteration networks. In Advances in Neural Information Processing Systems (pp. 2154-2162).
> > > > > > >
> > > > > > > [7] Neu, G. and Pike-Burke, C., 2020. A unifying view of optimism in episodic reinforcement learning. Advances in Neural Information Processing Systems, 33.

---

### Official Review · AnonReviewer5 · 2020-11-11

**Rating:** 4
**Confidence:** 4

**Review:**

Summary:

The paper claims to study the sample efficiency in policy optimization with general function approximation. Specifically, the authors propose an algorithm with transition model approximation and analyze the regret when adopting RKHS or function classes with bounded Eluder dimension. The results also apply to neural networks in the NTK regime.

-----------------------------------------------------------------------------------------

Reasons for score:

The theoretical understanding of RL with function approximation is an important issue. It is nice to see that the paper provides a wide-ranging discussion under this topic. My major concerns are about the novelty and clarity. The paper investigates many interesting scenarios, but disappointingly, none of them is satisfyingly addressed.

-----------------------------------------------------------------------------------------

Pros:
* The paper goes beyond tabular MDP and finite-dimensional linear function setting. The use of RKHS and Eluder dimension makes the results more general.
* The paper connects reinforcement learning with neural network and attempts to provide a unified theoretical explanation for the empirical successes in deep RL.

-----------------------------------------------------------------------------------------

Cons:
* In introduction, the authors claim to study a policy-based approach but their algorithm turns out to be a model-based one.
* The paper seems only a combination of Cai et al. (2019) and Ayoub et at. (2020) using RKHS and Eluder dimension techniques. Admittedly, the extension needs some work, however, the results do not provide much more insights compared with Cai et al. (2019).
* Assumption 4.2 is too restrictive. It assumes the eigenvalues of kernel to decay exponentially fast. A discussion of power law spectral decay would definitely improve the quality of the paper. In Assumption 4.6, it is also unclear why exponential spectral decay is a reasonable assumption on NTK.
The authors refer to Srinivas et al. (2019) and Yang & Salman (2019) to justify their assumptions. However, it seems to the reviewer that Srinivas et al. (2019) considers all finite spectrum, exponential spectral decay and power law spectral decay. Also, Theorem 3.1 in Yang & Salman (2019) shows a power law for NTK and does not support Assumption 4.6.

---

> ### Author Response · Authors · 2020-11-19
> **Reply to Reviewer 5**
>
> We appreciate the valuable review and suggestions. We have revised our work accordingly.
>
> In what follows, we address your concerns in detail.
>
> 1.(policy-based approach) The algorithm of the paper is model-based, and we have made it clear in the abstract and introduction. However, it seems that the reviewer wrongly claims that an RL algorithm cannot be both model-based and policy-based. We would like to point out that an RL method is model-based or model-free only depends on whether the algorithm estimates the transition or not. While an algorithm is policy-based if it directly updates the policy via optimization methods where the update direction can be computed by either model-based or model-free methods. Thus, they are orthogonal aspects of an RL algorithm, and an algorithm can be simultaneously model-based and policy-based. This is exactly the case of our algorithm --- we estimate the model via value-targeted regression, use the estimated model to estimate the critic, and update the policy by a KL-divergence based optimization problem based on the estimated critic.
>
>
> 2.(Novelty) Our paper is NOT simply a combination of [1] and [2].
>
> Compared with [2], we for the first time introduce RKHS (and the NTK regime of overparameterized neural networks) into the eluder framework, which is not studied before even in bandit. [2] and other previous work on the eluder framework only analyzes the eluder dimension of finite-dimensional linear models or generalized finite-dimensional linear models with very limited link functions. Bringing infinite-dimensional RKHS and overparameterized neural networks require nontrivial efforts.
>
> Moreover, in terms of algorithms, [2] propose a UCRL-type algorithm whereas we study policy optimization algorithm. These two algorithms have diverse properties -– the value functions constructed by [2] are always upper bounds on the globally optimal value function $Q^{\pi*}$ whereas the value functions constructed by OPPO are upper bounds for policy evaluation problems, which can be much smaller than $Q^{\pi*}$.
>
> In addition, [1] only studies the linear function approximation setting and both the algorithm and theory are specific to such a linear setting. In comparison, we extend its algorithm to the general function approximation setting which includes linear, RKHS, overparameterized neural networks as a special case. We emphasize that the algorithm in [1] cannot be directly applied to the general function approximation setting. In addition to the novel algorithm, we also establish a unified theoretical framework that covers linear, RKHS, and overparameterized neural networks as special cases.
>
> Finally, we emphasize that our result cannot be directly obtained by combining [1] and [2] for two reasons. First, RKHS and overparameterized neural network cases are not covered in [1] or [2]. Second, the algorithm in [2] cannot be directly combined with [1]. Thus, we propose a novel optimistic policy optimization problem to tackle such a challenge. Specifically, the optimistic policy evaluation subroutine is novel and has not appeared in [1] or [2]. To the best of our knowledge, our work establishes for the first time a policy optimization algorithm that provably explores under the general function approximation setting.
>
> To analyze this algorithm, we provide a general theorey framework that bridges policy optimization and general function approximation under the eluder dimension framework and establishes a regret bound. Our theory covers the result in [1] as a special example while also cover powerful function approximators such as RKHS functions and overparameterized neural networks. To this end, we further enrich the theory of the eluder dimension and demonstrate the power of our theory framework by establishing the eluder dimension of RKHS, and overparametrized neural networks.
>
> Furthermore, in terms of technical contribution, more importantly, our paper is the first that incorporates RKHS (and the NTK regime of overparametrized neural networks) into the eluder framework, which is not studied before even for bandits. Such a novel eluder theory is not only limited to policy optimization but also can contribute to the theoretical study of other RL algorithms with kernel or neural network function approximation including UCRL [2] and model-based value iteration.

---

> > ### Author Response · Authors · 2020-11-19
> > **Reply to Reviewer 5 - continued**
> >
> > 3.(exponential decay) Kernels with exponentially decaying eigenvalues widely exist. We have provided concrete examples of kernels with exponentially decaying eigenvalues in the revised pdf. In particular, we show that (1) the squared exponential kernel has exponentially decaying eigenvalues; (2) the population neural tangent kernel of a two-layer neural network has exponentially decaying eigenvalues when the activation function is the sine function. See Section G in the appendix for more details.
> >
> > Furthermore, we remark that our upper bound on the eluder dimension of RKHS is not limited to exponential decay. Similar to [3], our proof can cover kernels with finite-rank, exponential decay, and polynomial decay eigenvalue conditions. Specifically, in the proof of Lemma C.1 in Appendix C, our proof for the exponential decay case can be made general. For finite-rank case, we can let $d_0$ be the rank of the kernel, i.e., the number of nonzero eigenvalues. Then, the same proof still holds and we can show that the eluder dimension is bounded by $ \mathcal{O} (d_0 \log (RKH))$.  Moreover, for the polynomial decay case, we can still truncate at $d_0$ and calculate $\Gamma(d_0)$ in (C.4) using the polynomial eigenvalue decay condition. It can be shown that $\Gamma(d_0)$ decays polynomially in $d_0$. Then, we can find a proper $d_0$ by solving (C.12) and follow the same proof afterward. This yields an upper bound on the eluder dimension for the polynomial decay case. We remark that the ReLU NTK satisfies the polynomial eigenvalue decay condition [5].
> >
> >
> > Reference
> >
> > [1] Cai, Q., Yang, Z., Jin, C. and Wang, Z., 2019. Provably efficient exploration in policy optimization. arXiv preprint arXiv:1912.05830.
> >
> > [2] Ayoub, A., Jia, Z., Szepesvari, C., Wang, M. and Yang, L.F., 2020. Model-Based Reinforcement Learning with Value-Targeted Regression. arXiv preprint arXiv:2006.01107.
> >
> > [3] Srinivas, N., Krause, A., Kakade, S.M. and Seeger, M., 2009. Gaussian process optimization in the bandit setting: No regret and experimental design. arXiv preprint arXiv:0912.3995.
> >
> > [4] Yang, Z., Jin, C., Wang, Z., Mengdi, W., and Jordan, I.M., 2020. Bridging Exploration and General Function Approximation in Reinforcement Learning: Provably Efficient Kernel and Neural Value Iterations. arXiv preprint arXiv:2011.04622.
> >
> > [5] Wang, W., Hu, T., Lin, C. and Cheng, G., 2020. Regularization Matters: A Nonparametric Perspective on Overparametrized Neural Network. arXiv preprint arXiv:2007.02486.

---

### Author Response · Authors · 2020-11-23
**General Responses**

We thank all four reviewers for helpful and valuable feedback. We have made adjustments to the main paper and appendix accordingly.

We would like to highlight that our paper is novel in both the algorithm and theory. Compared with the previous work [1] studying linear models, we provide an optimistic policy optimization algorithm framework with general function approximations on the transition kernel and establish the regret upper bound in the case of RKHS and overparametrized neural networks. We note that extending the algorithm to the general function approximation setting requires a modification of the optimistic policy evaluation subroutine. The algorithm in [1] cannot be directly applied to the setting of neural networks as they are a class of nonlinear functions. Besides, directly applying the theory to the RKHS setting would lead to a vacuous bound as the dimensionality is infinity.

Moreover, our result **firstly** introduces the eluder dimension framework (bandit [2] and MDPs [3]) into policy optimization approaches and for the first time characterizes the eluder dimension of the infinite-dimensional functional classes including RKHS and overparameterized neural networks. Our novel eluder dimension results also contribute to other exploration algorithms in reinforcement learning. Moreover, we note that the algorithm in [3] is a UCRL-type algorithm that requires solving optimistic MDPs and obtaining upper bounds of the optimal value function. Thus, the algorithm in [3] cannot be directly modified into a policy optimization method. Thus, given a policy optimization algorithm for the linear function setting ([1]) and a UCRL type algorithm for the general function approximation setting ([3]), developing the algorithm and theory for policy optimization under the **general function approximation** setting including RKHS and neural networks remains **open**. Our paper address provides an affirmative answer to such an open problem.

We notice that there are some unfortunate misunderstandings in the setting, novelty, and contribution of our paper. We have addressed these misunderstandings and other questions in responses in detail and we sincerely hope that the paper could be (re-)evaluated with our explanations in mind. In particular, the setting of full reward function information is practical (See Point 1 to Reviewer 4), the algorithm is simultaneously model-based and policy-based (See Point 2 to Reviewer 4 and Point 1 to Reviewer 5), the algorithm and theory are strict generalizations of prior works rather than simple combinations (See Point 4 to Reviewer 4 and Point 2 to Reviewer 5), and the computation complexity is acceptable in the online learning setting (See Point 5 to Reviewer 4).

We have emphasized in the abstract and introduction that the algorithm is model-based. Moreover, we have added a section (Section G in appendix) to provide examples of kernel and NTK that satisfying the assumptions in the paper and explained how to extend our result to kernels with more general eigenvalue conditions.

[1] Cai, Q., Yang, Z., Jin, C. and Wang, Z., 2019. Provably efficient exploration in policy optimization. arXiv preprint arXiv:1912.05830.

[2] Russo, D. and Van Roy, B., 2013. Eluder dimension and the sample complexity of optimistic exploration. In Advances in Neural Information Processing Systems (pp. 2256-2264).

[3] Ayoub, A., Jia, Z., Szepesvari, C., Wang, M. and Yang, L.F., 2020. Model-Based Reinforcement Learning with Value-Targeted Regression. arXiv preprint arXiv:2006.01107.

---

### Decision · Program_Chairs · 2021-01-07
**Final Decision**

**Decision:**

Reject

**Comment:**

While the reviewers appreciated the aim of the work, they found the technical contribution to be too incremental to be of sufficient interest and the exploration of the problem and its significance to be incomplete in the paper's current state.